# MEMORY GYM: PARTIALLY OBSERVABLE CHALLENGES TO MEMORY-BASED AGENTS

**Marco Pleines**[1]    **Matthias Pallasch**[1]    **Frank Zimmer**[2]    **Mike Preuss**[3]
[1]TU Dortmund University    [2]Rhine-Waal University of Applied Sciences
[3]LIACS Universiteit Leiden
`marco.pleines@tu-dortmund.de`

## ABSTRACT

Memory Gym is a novel benchmark for challenging Deep Reinforcement Learning agents to memorize events across long sequences, be robust to noise, and generalize. It consists of the partially observable 2D and discrete control environments Mortar Mayhem, Mystery Path, and Searing Spotlights. These environments are believed to be unsolvable by memory-less agents because they feature strong dependencies on memory and frequent agent-memory interactions. Empirical results based on Proximal Policy Optimization (PPO) and Gated Recurrent Unit (GRU) underline the strong memory dependency of the contributed environments. The hardness of these environments can be smoothly scaled, while different levels of difficulty (some of them unsolved yet) emerge for Mortar Mayhem and Mystery Path. Surprisingly, Searing Spotlights poses a tremendous challenge to GRU-PPO, which remains an open puzzle. Even though the randomly moving spotlights reveal parts of the environment's ground truth, environmental ablations hint that these pose a severe perturbation to agents that leverage recurrent model architectures as their memory. Source Code: `https://github.com/MarcoMeter/drl-memory-gym/`

## 1 INTRODUCTION

Memory is a vital mechanism of intelligent living beings to make favorable decisions sequentially under imperfect information and uncertainty. One's immediate sensory perception may not suffice if information from past events cannot be recalled. Reasoning, imagination, planning, and learning are skills that may become unattainable. When developing autonomously learning decision-making agents, the agent's memory mechanism is required to maintain a representation of former observations to ground its next decision. Adding memory mechanisms as a recurrent neural network (Werbos, 1990) or a transformer (Vaswani et al., 2017) led to successfully learned policies in both virtual and real-world tasks. For instance, Deep Reinforcement Learning (DRL) methods master complex video games such as StarCraft II (Vinyals et al., 2019), and DotA 2 (Berner et al., 2019). Examples of successes in real-world problems are dexterous in-hand manipulation (Andrychowicz et al., 2020) and controlling tokamak plasmas (Degrave et al., 2022). In addition to leveraging memory, these tasks require vast amounts of computation resources and additional methods (e.g. domain randomization, incorporating domain knowledge, ect.), which make them undesirable for solely benchmarking an agent's ability to interact with its memory meaningfully.

We propose Memory Gym as a novel and open source benchmark consisting of three unique environments: Mortar Mayhem, Mystery Path, and Searing Spotlights. These environments challenge memory-based agents to memorize events across long sequences, generalize, be robust to noise, and be sample efficient. By accomplishing the desiderata that we define in this work, we believe that Memory Gym has the potential to complement existing benchmarks and therefore accelerate the development of DRL agents leveraging memory. All three environments feature visual observations, discrete action spaces, and are notably not solvable without memory. This allows users to assure early on whether their developed memory mechanism is working or not. To fit the problem of sequential decision-making, agents have to frequently leverage their memory to solve the posed tasks by Memory Gym. Several related environments ask the agent only to memorize initial cues, which require infrequent agent-memory interactions. Our environments are smoothly configurable. This

is useful to adapt the environment's difficulty to the available resources, while easier difficulties can be used as a proof of concept. Competent methods can show off themselves in new challenges or identify their limits in a profound way. All environments are procedurally generated to evaluate the agent's ability to generalize to unseen levels (or seeds). Due to the aforementioned configurable difficulties, the trained agent can be evaluated on out-of-distribution levels. Memory Gym's significant dependencies are gym (Brockman et al., 2016) and PyGame[1]. This allows Memory Gym to be easily set up and executed on Linux, macOS, and Windows. Multiple thousands of agent-environment interactions per second are simulated by all environments.

This paper proceeds as follows. We first define the memory benchmarks' criteria and portray the related benchmarks' landscape. Next, Mortar Mayhem, Mystery Path, and Searing Spotlights are detailed. Afterward, we show that memory is crucial in our environments by conducting empirical experiments using a recurrent implementation (GRU-PPO) of Proximal Policy Optimization (PPO) (Schulman et al., 2017) and HELM (Paischer et al., 2022). When scaling the hardness in Mortar Mayhem and Mystery Path, a full range of difficulty levels emerge. Searing Spotlights remains unsolved because the recurrent agent is volatile to perturbations of the environment's core mechanic: the randomly wandering spotlights. This observation is also apparent when training on a single Procgen BossFight (Cobbe et al., 2020) level under the same spotlight perturbations as in Searing Spotlights. At last, this work concludes and enumerates future work.

## 2    COMPARISON OF RELATED MEMORY BENCHMARKS

### 2.1    DESIDERATA OF MEMORY BENCHMARKS

Before detailing related benchmarks, we define the aforementioned desiderata that we believe are essential to memory benchmarks and benchmarks in general.

**Accessibility** refers to the competence to easily set up and execute the environment. Benchmarks shall be publicly available, acceptably documented, and open source while running on the commonly used operating systems Linux, macOS, and Windows. Linux, in general, is important because many high-performance computing (HPC) facilities employ this operating system. As HPC facilities might not support virtualization, benchmarks should not be solely deployed as a docker image or similar. At last, benchmarks shall run headless because otherwise, these potentially rely on dependencies like xvfb or EGL, which HPC facilities may not support as well. Suppose relevant benchmark details, such as environment dynamics, are missing. In that case, it can be desirable to support humanly playable environments so that these can be explored in advance.

**Fast simulation speeds**, which achieve thousands of steps per second (i.e. FPS), allow training runs to be more wall time efficient, enabling to upscale experiments and their repetitions or faster development iterations. The benchmark's speed also depends on the time it takes to reset the environment to set up a new episode for the agent to play. Towards maxing out FPS on a single machine, benchmarks shall be able to run multiple instances of their environments concurrently.

**High diversity** attributes environments that offer a large distribution of procedurally generated levels to reasonably challenge an agent's ability to generalize (Cobbe et al., 2020). Also, it is desirable to implement smoothly configurable environments to evaluate the agent at out-of-distribution levels.

**Scalable difficulty** is a property that shall make environments controllable such that their current hardness can be increased or decreased. Easier environments can have benefits: a proof-of-concept state is sooner reachable while developing novel methods, and research groups can fit the difficulty to their available resources. Moreover, increasing the difficulty ensures that already competent methods may prove themselves in new challenges to demonstrate their abilities or limits.

**Strong dependency on memory** refers to tasks that are only solvable if the agent can recall past information (i.e. successfully leveraging its memory). Section 2.4 describes partially observable environments that can be solved to some extent without memory. While memory-based agents might more efficiently solve these tasks, these do not guarantee that the agent's memory is working. To ensure that the utilized memory mechanism is working and does not suffer from bugs, this criterion cannot be omitted by benchmarks targeting specifically the agent's memory.

---

[1] https://www.pygame.org

Table 1: An overview of the simulation speed and meta desiderata of the considered environments. The meta desiderata cover the criteria that are applicable to any benchmark in DRL. The mean FPS are measured across 100 episodes using constant actions. Procgen is averaged across its memory distribution environments, and so is Memory Gym. The other benchmarks are measured using a single environment. ● refers to true and ○ to false. Appendix B provides further details.

| Benchmark | Mean FPS | Publicly Available | Open Source | Linux | macOS | Windows | Headless | Docker | Concurrent | Playable | High Diversity | Scalable Difficulty |
|---|---|---|---|---|---|---|---|---|---|---|---|---|
| **Memory Gym (ours)** | 10123 | ● | ● | ● | ● | ● | ● | ○ | ● | ● | ● | ● |
| Procgen Memory Distribution (Cobbe et al., 2020) | 18530 | ● | ● | ● | ● | ● | ● | ○ | ● | ● | ● | ● |
| DM Ballet (Lampinen et al., 2021) | 6631 | ● | ● | ● | ● | ● | ● | ○ | ● | ● | ● | ● |
| MiniGrid Memory (Chevalier-Boisvert et al., 2018) | 5185 | ● | ● | ● | ● | ● | ● | ○ | ● | ● | ○ | ● |
| VizDoom (Wydmuch et al., 2018) | 549 | ● | ● | ● | ● | ● | ● | ○ | ● | ● | ● | ● |
| DM Memory Task Suite (Fortunato et al., 2019) | 442 | ● | ○ | ● | ● | ● | ● | ○ | ● | ● | ● | ● |
| DM Lab 30 (Beattie et al., 2016) | 433 | ● | ● | ● | ○ | ○ | ○ | ○ | ○ | ● | ● | ● |
| DM Numpad (Parisotto et al., 2020) | | ○ | | | | | | | | | | |
| DM Memory Maze (Parisotto et al., 2020) | | ○ | | | | | | | | | | |
| DM Object Permanence (Lampinen et al., 2021) | | ○ | | | | | | | | | | |

**Strong dependency on frequent memory interactions** is a property of tasks that forces the agent to recall information from and add information to its memory frequently. We believe this is more suitable for sequential decision-making problems because some related environments can be turned into supervised learning problems (Section 2.5) and therefore only assess the memory's capacity and potentially its robustness to noise.

## 2.2 CONSIDERED ENVIRONMENTS

A diverse set of environments were used in the past to challenge memory-based agents. Some of them are originally fully observable but are turned into partially observable Markov Decision Processes (POMDP) by adding noise or masking out information from the agent's observation space. For instance, this was done for the Arcade Learning Environment (Bellemare et al., 2013) by using flickering frames (Hausknecht & Stone, 2015) and common control tasks by removing the velocity from the agent's observation (Heess et al., 2015; Meng et al., 2021; Shang et al., 2021). These environments do not require the agent to memorize long sequences and can already be approached using frame stacking. Control tasks also touch on the context of Meta Reinforcement Learning (Meta RL), where memory mechanisms are prominent (Wang et al., 2021; Melo, 2022; Ni et al., 2022). The same applies to Multi-Agent Reinforcement Learning (Berner et al., 2019; Baker et al., 2020; Vinyals et al., 2019). As we solely focus on benchmarking the agent's memory and its ability to generalize, we do not compare Memory Gym to environments of more complex contexts such as DM Alchemy (Wang et al., 2021), Crafter (Hafner, 2021), or Obstacle Tower Juliani et al. (2019). Those might need additional components to the agent's architecture and its training paradigm. Within this section, we consider DRL benchmarks (Table 1) that were used by the recently contributed memory approaches MRA (Fortunato et al., 2019), GTrXL (Parisotto et al., 2020), HCAM (Lampinen et al., 2021), HELM (Paischer et al., 2022), and A2C-Transformer (Sopov & Makarov, 2022). By examining these works, it becomes apparent that all of them use different environments to evaluate their methods, making their results harder to compare.

## 2.3 META DESIDERATA

Table 1 provides information on the meta desiderata of the considered environments. Some environments are inaccessible to some extent, as some cannot be run headless. DM Memory Maze (Parisotto et al., 2020), DM Numpad (Humplik et al., 2019), and DM Object Permanence (Lampinen et al., 2021) are not publicly available and can, therefore, not be used to reproduce and compare results in adjacent research. DM Memory Task Suite (Fortunato et al., 2019) is not open source and can

only be accessed via docker. It may be for these reasons that DM Memory Task Suite, Object Permanence, Memory Maze, and Numpad are not widely used yet. DM Lab 30 (Beattie et al., 2016) has more uses, but the available environments, especially those used in the context of memory, are superficially documented. Parisotto et al. 2020 divide these environments into reactive and memory tasks without further reasoning. Potential users have to explore these environments by playing them, which is possible on machines running Linux only. Procgen (Cobbe et al., 2020), DM Ballet (Lampinen et al., 2021), MiniGrid (Chevalier-Boisvert et al., 2018), and Memory Gym (ours) are the only environments that achieve a throughput of thousands of steps per second. 3D environments usually achieve a smaller throughput, like VizDoom (Wydmuch et al., 2018) or DM Memory Task Suite (Fortunato et al., 2019).

### 2.4 Strong Dependency on Memory

As stated before, if an environment is solvable to some extent using an agent without memory, it is not easy to differentiate whether the memory mechanism is working. This impression can be retrieved by examining the results of Sopov & Makarov 2022 on the VizDoom environments where a policy without memory makes progress or achieves comparable performances. The six environments of Procgen's memory distribution leave us with controversial thoughts. Paischer et al. 2022 show results in these environments that indicate that a memory-less approach can perform similarly to a memory one. We further take a closer look at Miner. The agent can momentarily perceive some cues related to past steps of the episode: its last action is retrievable from its rotation. At the same time, the density of vanished tiles shows the agent whether this region was explored or not. Exploiting these cues potentially helps memory-less agents to find successful strategies and make us believe that this environment does not strongly depend on memory, while doubtlessly, a memory-based agent should be more efficient concerning the number of steps needed to solve the entire task. Another problem is that the mean cumulative reward is usually reported, which averages the rewards achieved by collecting diamonds and using the exit. This data does not precisely tell how good an agent is at completing the entire task. Especially recalling the exit's position seems crucial to succeeding as soon as possible. Reporting the success rate and the episode length should provide more meaningful insights. The environments Heist and Maze encompass fully observable levels, which raises concerns on strong memory dependence.

### 2.5 Strong Dependency on Frequent Memory Interactions

Concerning frequent memory interactions, we believe that MiniGrid Memory (Chevalier-Boisvert et al., 2018), Spot the Difference (Fortunato et al., 2019), and DM Ballet (Lampinen et al., 2021) are not well suited to the need for memory in sequential decision making. These environments demand the agent's memory to solely memorize the to-be-observed goal cues during the very beginning of the episode. Once this cue is memorized, there is no need to manipulate the agent's memory further. Simply maintaining it is enough to solve the task. It can be hypothesized that the extracted features from observing the goal cues are sufficient to solve the ballet environment. To show this, the ballet environment can be made fully observable (markovization) by feeding the entire sequence of cues to a recurrent encoder that extracts features that are utilized by the agent's policy. Consequently, the policy does not need to maintain its memory in this case. The task at hand gets much easier because the agent does not need to make obsolete decisions while observing the sequence of cues as its position is frozen. These tasks would, therefore, only challenge the capacity and robustness to noise of the agent's memory. We believe this can be done more efficiently by other benchmarks that do not belong to the context of DRL, like Long Range Arena (Tay et al., 2021).

## 3 Memory Gym Environments

Mini-games of the commercial video game Pummel Party[2] inspired Memory Gym's environments. The agent perceives all environments using $84 \times 84$ RGB pixels, while its action space is multi-discrete, featuring two dimensions of size three as shown in appendix C.1. One dimension allows the agent to move horizontally (no movement, move left, move right), and the other concerns the agent's vertical locomotion (no movement, move up, move down). Mortar Mayhem and Mystery

---

[2]http://rebuiltgames.com/

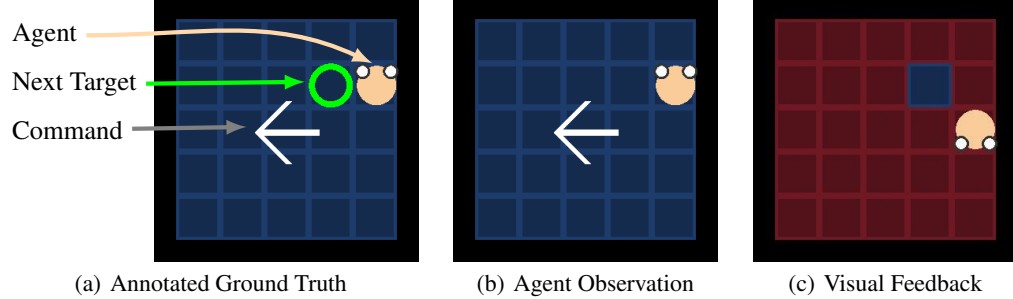

|                    |                    |                    |
| ------------------ | ------------------ | ------------------ |
| (a) Annotated Ground Truth | (b) Agent Observation | (c) Visual Feedback |

Figure 1: Mortar Mayhem's relevant entities are presented by the annotated ground truth of the environment (a). The green circle, which is solely part of the ground truth, indicates the very next tile to move to. Once the agent has solved this command, the green circle moves to the next target tile. At the start of an episode, the commands are rendered sequentially onto the agent's observation (b) while the agent cannot move. Once all commands are shown, the agent has to move to the target tile in a certain amount of time. As soon as the time for executing the current command is over, the agent's success is verified, as seen in figure (c). This visual feedback is perceivable by the agent. After a delay of a few steps, the agent can approach the following command if the episode did not terminate yet due to failure or completing the entire command sequence.

Path also feature variants based on grid-like locomotion (Figures 10(b) to 10(d)). This way, the action space is discrete and allows the agent to not move at all, rotate left, rotate right, or move forward. Subsequently, we further detail the environments' dynamics, their peculiarities towards memory, and how these can be smoothly scaled to support varying difficulty levels. Appendix C.2 quantifies the episode lengths. All parameters used for scaling the environments' hardness and their default values are found in appendix C.3. Appendices C.4 to C.6 visualize played episodes.

## 3.1 MORTAR MAYHEM

Mortar Mayhem (MM) (Figure 1) takes place inside a grid-like arena and consists of two tasks that depend on memory. At first, while unable to move, the agent has to memorize a sequence of five commands (Clue Task), and afterward, it has to execute each command in the observed order (Act Task). One command orders the agent to move to one adjacent floor tile or to stay at the current one. If the agent fails, the episode terminates while receiving no reward. Upon successfully executing one command, the agent receives a reward of $+0.1$. Episode lengths in MM are dependent on the agent's current ability. The better the agent, the longer the episode lasts until an upper bound (i.e. max episode length) is reached. MM can be reduced to provide only the Act Task (MMAct). In this case, the command sequence is fully observable as a one-hot encoded feature vector. The Act Task requires an agent to leverage its memory frequently because otherwise, the agent does not know which commands were already executed to fulfill the next one, while there are nine tiles at maximum to consider. To solve this problem, the agent could learn to track time (e.g. count steps) where a short-term memory should suffice. The hardness of MM can be further simplified by equipping the agent with a grid-like movement (MMGrid, MMActGrid). The agent is now capable of moving one tile at a time. To ensure a meaningful task, the agent must execute ten commands, not five. Further examples to raise MM's difficulty are to extend or sample the number of commands or the delay between command execution. If compared to DM Ballet, MM's Act Task requires many correct actions in sequence, while DM Ballet asks the agent to only identify the requested dancer.

## 3.2 MYSTERY PATH

Mystery Path (MP) (Figure 2) challenges the agent to traverse an invisible path in a grid-like level of dimension $7 \times 7$, while only the path's origin is visible to the agent. If the agent moves off the path (i.e. falls down the pit), the agent is relocated to its origin. The episode terminates if the agent reaches the goal or runs out of time (512 steps). Upon reaching the goal, the environment signals a reward of $+1$. To overcome uncertainty in this environment, the agent has to memorize several locations: its steps on the path and the locations where it fell off. The invisible path is procedurally generated using the path-finding algorithm $A^*$ (Hart et al., 1968). At first, the path's

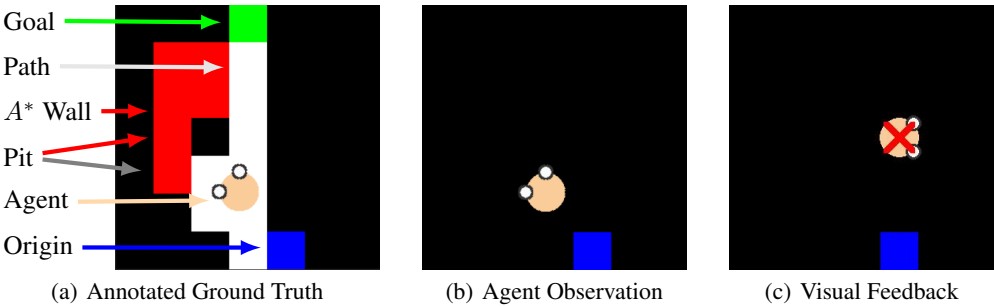

Figure 2: Mystery Path's relevant entities are described by the environment's annotated ground truth (a). The walkable parts of the path are colored blue (origin), green (goal), and white (path). Every other part of the environment is considered a pit (black and red) where the agent falls off and is then relocated to the path's origin during its next step. The initially randomly chosen red tiles ($A^*$ Walls) are not considered during path generation. The agent observes itself and the path's origin (b). If the agent is off the path, a red cross provides visual feedback (c), which is also observed.

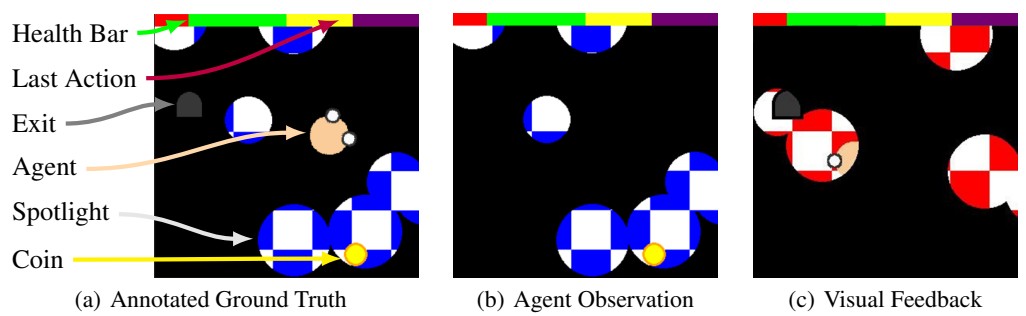

Figure 3: All relevant entities can be identified in the annotated ground truth of Searing Spotlights (a). The top rows of pixels feature the agent's remaining health points and its last action that two chunks and three colors encode. Yellow circles correspond to coins, while the somewhat rounded gray shape resembles the exit. If the exit is open (i.e. no coins are left), it turns green. The floor of the environment is a chessboard colored blue and white. As seen in the agent's observation (b), the spotlights hide or reveal the other entities. As additional visual feedback (c), the blue floor tiles turn red if a spotlight spots the agent.

origin is sampled from the grid's border. Then, the goal is placed in a random position on the opposing side. Usually, $A^*$ seeks to find the shortest path. $A^*$'s cost function is sampled to ensure that a wide variety of paths is procedurally generated. The path generation randomizes some cells to be untraceable ($A^*$ walls). Note that the walls are considered pits and not physical obstacles. MP can also be simplified to a grid variant (MPGrid), featuring the single discrete action space of four actions, while the agent's time limit is reduced from 512 steps to 128. In contrast to MM, the episode terminates sooner as the agent improves.

## 3.3 SEARING SPOTLIGHTS

Searing Spotlights (Figure 3) is perceived as a pitch-black surrounding by the agent where the only information is unveiled by roaming and threatening spotlights. While initially starting with a limited number of health points (e.g. 100), the agent loses one health point per step if hit by a spotlight. The episode terminates if the agent has no remaining health points or runs out of time (512 steps). Because of the posed threats, the agent has to hide in the dark. To successfully run away from closing in spotlights, the agent must memorize its past actions and at least one past position of itself to infer its current location. That also requires the agent to carefully use its health point budget to figure out its location - ideally once briefly. To avoid numb policies, two additional tasks are added to

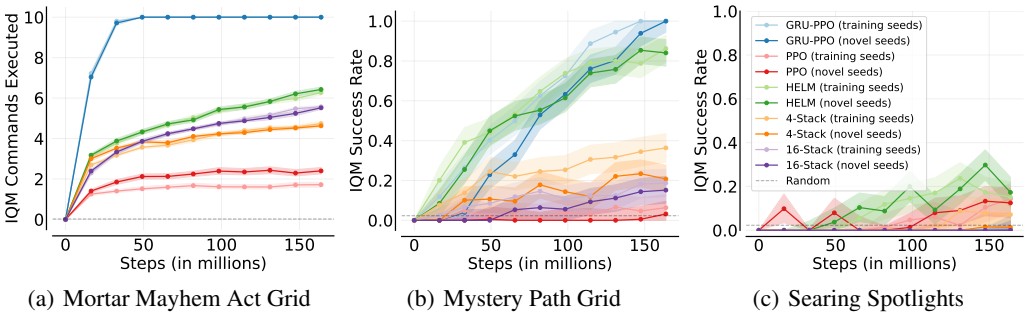

Figure 4: Training and generalization performances show strong dependence on memory.

the environment that requires the agent to traverse the environment. The first one randomly places an exit in the environment, which the agent can use to receive a reward of $+1$ and terminate the episode. Secondly, a coin collection task is added. Before using the environment's exit, the agent has to collect a certain number of randomly positioned coins. Collecting one coin grants a reward of $+0.25$ to the agent. If the agent successfully uses its memory, it can infer its current location, and recall the locations of the exit and the coins. A simplified grid version of Searing Spotlights is not available. As a default simplification, the episode starts with perfect information, and after a few steps, the global light is dimmed until off. Just like in MP, successful episodes terminate sooner. Appendix C.3 enumerates multiple parameters that organize the measures, behavior, and spawning frequency of the spotlights. Other parameters concern the reward function, the agent's properties, and the visual representation of Searing Spotlights.

## 4 BASELINE EXPERIMENTS

We run empirical baseline experiments using the DRL algorithm Proximal Policy Optimization (PPO) (Schulman et al., 2017). To support memory, our implementation adds a gated recurrent unit (GRU) (Cho et al., 2014) to the actor-critic model architecture of PPO (GRU-PPO). Related memory-based approaches such as MRA (Fortunato et al., 2019), GTrXL (Parisotto et al., 2020), HCAM (Lampinen et al., 2021), and ALD (Parisotto & Salakhutdinov, 2021) are not considered, because there is no applicable code available (see appendix G), while these methods are expensive to reproduce. That does not account for HELM (Paischer et al., 2022), which we successfully added to our training framework. However, HELM has poor wall-time efficiency (see figs. 17 and 18 in appendix E). Due to the vast transformer component consisting of 18 blocks, the policy takes much more time to produce actions during inference. One GRU-PPO training run takes about ten hours in MMActGrid with a sequence length of 79, while HELM needs at least six times longer. This effect worsens for longer sequences. The last two baselines consider frame stacking. One stacks 4 RGB frames (4-Stack), while the other one stacks 16 grayscale frames (16-Stack). We repeat all experiments 5 times. All training runs utilize 100,000 environment seeds. Generalization is assessed on 30 novel seeds, which are repeated 5 times. Hence, each data point aggregates 750 episodes. The subsequent sample efficiency curves show the interquartile mean (IQM) and a confidence interval of 0.95 as recommended by Agarwal et al. (2021). For Mystery Path and Searing Spotlights, we report the success rate, which indicates whether the agent succeeded at the entire task or not. Results on Mortar Mayhem show the number of commands that were properly executed during one episode. Appendix D details all baselines, the model architecture, and the used hyperparameters.

### 4.1 DEPENDENCY ON MEMORY

To support our claim that Memory Gym's environments strongly depend on memory, the agents are trained with PPO (memory-less), GRU-PPO, frame stacking, and HELM on MMActGrid, MPGrid (hidden goal), and Searing Spotlights. Figure 4(a) shows the results on MMActGrid where GRU-PPO successfully executes ten commands after 50 million training steps, while memory-less PPO is ineffective by solving only one to two commands. Even though the task requires short-term memory only, HELM and the frame stacking baselines are notably inferior to GRU-PPO. Concerning MPGrid

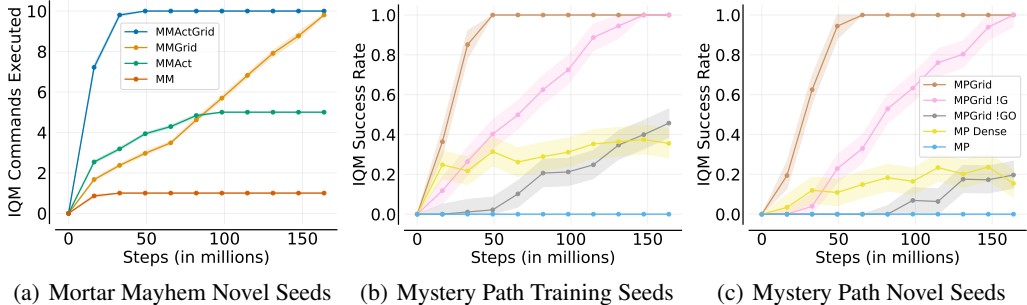

(a) Mortar Mayhem Novel Seeds    (b) Mystery Path Training Seeds    (c) Mystery Path Novel Seeds

Figure 5: By scaling the hardness of Mortar Mayhem and Mystery Path, unsolved and diverse levels of difficulties emerge. The agents are trained using GRU-PPO. MMAct and MM ask for five commands to be executed and not ten. Only the generalization performance is shown because, as in figure 4(a), both performances are nearly identical. In MPGrid, the goal and origin are visible, while the other two experiments hide the goal (!G) or both (!GO). MP Dense provides a denser reward signal. Whenever the agent touches a tile of the path for the first time, it is rewarded by 0.03.

(Figure 4(b)), GRU-PPO needs the entire training time to succeed and generalize. HELM reaches a success rate of about 0.84 during generalization. The frame stacking agents perform worse, while PPO's performance is close to random. Results of Searing Spotlights are illustrated in Figure 4(c). All baselines are barely able to reliably complete this environment, although HELM is the best baseline achieving a success rate of about 0.3. To further investigate this outcome, subsection 4.3 examines environment ablations and hints at why randomly moving spotlights may be a significant issue to memory based on recurrence.

## 4.2 LEVELS OF DIFFICULTY IN MORTAR MAYHEM AND MYSTERY PATH

Levels of difficulty, ranging from easy to unsolved, emerge when scaling Mortar Mayhem and Mystery Path and if trained using GRU-PPO. As seen in figure 5(a), the agent in MMActGrid only needs 50 million steps to accomplish the whole task. In MMAct, it needs twice as long to solve only five commands. Note that the available commands in the grid variants comprise only five command choices and not the full range of nine ones. Adding the Clue Task notably increases the agent's challenge because the agent trained on MMGrid needs the entire training time to succeed. At the same time, the complete task of MM remains unsolved. The performance on training and novel seeds is nearly identical with nearly no variance, which can be explained by the high uncertainty of this environment. A single observation leaves the agent clueless about which of the nine tiles to move next. Therefore the agent has no other choice but to obtain the ability to memorize and recall past events.

Concerning MPGrid, if the goal and origin are part of the agent's observation, it takes about 65 million steps for the agent to generalize to novel seeds, as shown in figure 5(c). If only the goal is hidden, the agent needs the entire training time to reach a success rate of 1.0. Hiding the goal and origin degrades the agent's performance to a success rate of 0.2 on novel seeds. Agents trained on MP or MP Dense do not evaluate well on novel seeds. The agent in MP Dense makes little progress due to the help of a denser reward function that signals a reward of 0.03 for visiting a tile of the path for only the first time. Consequently, we also experimented with a negative reward of $-0.001$ for each step taken, severely hurting the training performance.

## 4.3 RECURRENCE IS VULNERABLE TO SPOTLIGHT PERTURBATIONS

Searing Spotlights remains a puzzle. It is difficult to tell whether the agent's trained recurrent policy learned anything meaningful. Collecting the coin seems worse than chance. The agent's remaining health points drop to zero in nearly all evaluated episodes because avoiding the spotlights does not work well. It seems that the agent struggles to determine what it controls. Even though the spotlights unveil information on the ground truth of the environment or leave room for exploitation, the recurrent agent seems severely hurt by the perturbations inflicted by the randomly moving spotlights,

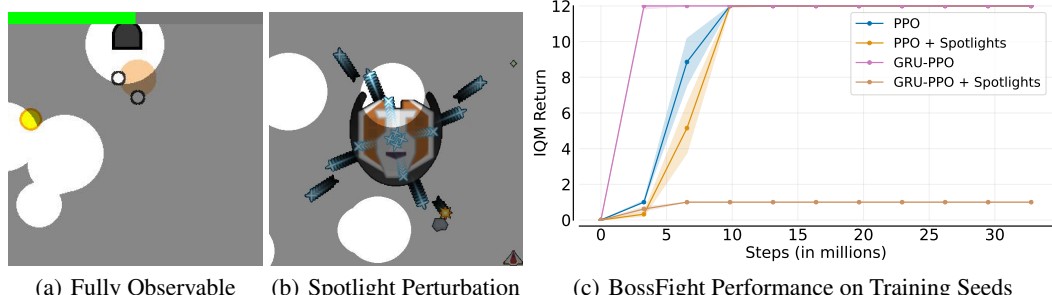

(a) Fully Observable     (b) Spotlight Perturbation     (c) BossFight Performance on Training Seeds

Figure 6: Under perfect information, the noise inflicted by spotlight perturbations harms the recurrent agents' performance in Searing Spotlights (a) and Procgen's BossFight (b). (c) denotes the training performances on training seeds when trained with and without perturbations using PPO and GRU-PPO. The IQM return denotes the summed rewards that the agent achieved during one episode.

which can be considered noise. Regardless of any executed measure (see appendix F for details), for instance, environmental ablations, model architectures, and hyperparameters, progress in learning a reasonable behavior is lacking. When trained with GRU-PPO, the only solvable scenario is to remove the spotlights. In this case, if the global light is initially turned on (perfect information) and dimmed until off during the first few steps, the agent rapidly collects the coin and uses the exit. In contradiction, if the global light is slightly dimmed while spotlights are present (Figure 6(a)), GRU-PPO fails. On the other hand, memory-less PPO quickly succeeds because this fully observable task is trivial.

Those results are also apparent in training a different environment. We choose Procgen's BossFight because it rather needs fewer steps to train as seen in the results of Cobbe et al. 2020. BossFight is set up to utilize only one level causing the boss always to behave the same. We extrapolate this level by 100,000 spotlight perturbation seeds under perfect information. Its background is rendered white to match Searing Spotlight's visual appearance more closely. The now-established BossFight task (Figure 6(b)) invites overfitting. Figure 6(c) shows results with and without spotlight perturbations on PPO and GRU-PPO. Notably, GRU-PPO, perturbed by spotlights, struggles.

## 5 CONCLUSION AND FUTURE WORK

By accomplishing the desiderata that we have defined and examined in this work, and by evaluating the baseline experiments, we believe that Memory Gym has proven potential to complement the landscape of Deep Reinforcement Learning benchmarks specializing on agents leveraging memory. Memory Gym's unique environments, Mortar Mayhem, Mystery Path, and Searing Spotlights, strongly depend on memory and frequent agent-memory interactions. As the environments' difficulties are smoothly scalable, current and future approaches to memory may find their right fit for examination. To our surprise, Searing Spotlights poses a yet unsolved challenge to agents leveraging GRU or LSTM.

In future work, it will be intriguing to see whether adjacent and novel memory approaches suffer from the noisy perturbations inflicted by the spotlights. HELM's performance on Searing Spotlights may point toward a transformer-based approach (e.g. HCAM (Lampinen et al., 2021), GTrXL (Parisotto et al., 2020)), while approaches that have not been used yet in the context of Deep Reinforcement Learning (e.g. Structural State-Space models (Gu et al., 2022)), are also considerable for Memory Gym in general. Consecutive work may extrapolate the concepts of Memory Gym's environments. Mortar Mayhem could be advanced to an open-ended environment. For instance, the Clue Task and the Act Task could be stacked infinitely. Every Clue Task could only show one command at a time, while the Act Task requires the agent to execute all commands that were shown so far. Hence, memory approaches could be compared by achieved episode lengths, which are correlated with the amount of information that the agent must be able to recall. Such a goal in mind may inspire Mystery Path as well. The to-be-traversed path could be endless. However, a larger environment may not fit into the agent's observation that an egocentric observation could compensate.

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

## A  INSTALLATION AND EXECUTION OF MEMORY GYM

```
# Create Anaconda environment
$ conda create -n memory-gym python=3.7 --yes
$ conda activate memory-gym
# Install Memory Gym
$ pip install memory-gym
# Play Mortar Mayhem
$ mortar_mayhem
```

Figure 7: Install and play Memory Gym's environments.

Memory Gym is available as a PyPi package[3], which installs the required dependencies gym and PyGame. We recommend to utilize Anaconda[4] to allow for console scripts to be executed. This way, the environments can be played using the following commands:

```
$ mortar_mayhem
$ mortar_mayhem_b
$ mortar_mayhem_grid
$ mortar_mayhem_b_grid
$ mystery_path
$ mystery_path_grid
$ searing_spotlights
```

Figure 8: Console scripts to play Memory Gym's environments

```
import gym
import memory_gym

# Configure desired reset parameters
options = {"allowed_commands": 5, "command_count": [5]}

env = gym.make("MortarMayhem-v0")
obs = env.reset(options)
print(obs["visual_observation"].shape) # (84, 84, 3)
print(env.action_space)                # MultiDiscrete([3 3])

done = False
while not done:
  action = env.action_space.sample()
  obs, reward, done, info = env.step(action)
```

Figure 9: Code interactions with Memory Gym environments

---

[3]Dear reviewers, the package will be available upon acceptance due to anonymity.
[4]https://www.anaconda.com/products/distribution

## B    EXTENDED RELATED BENCHMARK OVERVIEW

Table 2: Comparison of the simulation speed of our benchmark to related ones. A constant action is executed to measure the speed. Procgen encompasses the mean steps per second for the environments CaveFlyer, Jumper, Heist, Dodgeball, Maze, and Miner. The reset duration is included in steps per second. Some benchmarks are only measured by a single environment. The measurement was done on a AMD Ryzen 7 2700X.

| Benchmark | FPS | | Reset Duration |
| --- | --- | --- | --- |
| | Mean | Std | Mean (seconds) |
| Procgen Memory Distribution (Cobbe et al. (2020)) | 18530 | 5453 | 4e-6 |
| Mystery Path (ours) | 12187 | 330 | 5e-4 |
| Mortar Mayhem (ours) | 11692 | 179 | 2e-4 |
| Atari Breakout (Bellemare et al. (2013)) | 7117 | 19 | 6e-3 |
| DM Ballet (Lampinen et al. (2021)) | 6631 | 795 | 6e-4 |
| Searing Spotlights (ours) | 5490 | 69 | 5e-4 |
| MiniGrid Memory (Chevalier-Boisvert et al. (2018)) | 5185 | 141 | 1e-3 |
| MiniWorld TMaze (Chevalier-Boisvert et al. (2018)) | 1162 | 65 | 6e-4 |
| GridVerse MemoryNineRooms (Baisero & Katt (2021)) | 938 | 89 | 1e-3 |
| ML-Agents Hallway (Juliani et al. (2018)) | 702 | 42 | 1e-3 |
| VizDoom My Way Home (Wydmuch et al. (2018)) | 549 | 22 | 3e-3 |
| Crafter (Hafner (2021)) | 482 | 33 | 1e-1 |
| DM Spot the Difference (Fortunato et al. (2019)) | 442 | 20 | 1e-1 |
| DM Lab 30 rooms_watermaze (Beattie et al. (2016)) | 433 | 53 | 5e-1 |
| DM Alchemy (Wang et al. (2021)) | 308 | 31 | 2e-1 |
| DM Hard Eight ball room_navigation_cubes (Gülçehre et al., 2020) | 160 | 7 | 2e-1 |
| AnimalAI 3 aai_c32_r5 (Crosby et al. (2020)) | 190 | 59 | 1e-2 |
| Obstacle Tower (Juliani et al. (2019)) | 43 | 2 | 16e-1 |

# C  ENVIRONMENT DETAILS

## C.1  ACTION SPACES

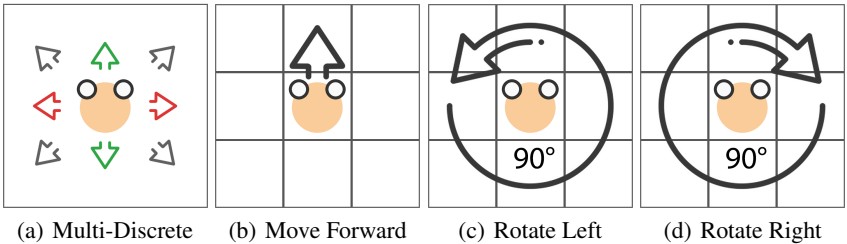

| (a) Multi-Discrete | (b) Move Forward | (c) Rotate Left | (d) Rotate Right |

Figure 10: Memory Gym's environments feature a multi-discrete action space as seen in (a). One dimension (green arrows) denotes the agent's vertical velocity. The other one (red arrows) refers to the agent's horizontal velocity. Both dimensions feature a no-op action. Therefore, the agent can utilize both dimensions to move into eight distinct cardinal directions (all arrows). The speed of the agent is fixed at 2.83 pixels per step. Mortar Mayhem and Mystery Path also provide a grid-like locomotion based on a discrete action space of four actions. The agent can move forward (b) one tile at a time. Its forward speed in Mortar Mayhem is 14 pixels per second. Its forward speed in Mystery Path is 12 pixels per second. Two other actions allow the agent to rotate left (c) or right (d) at 90 degrees. The last action is no-op.

## C.2  EPISODE LENGTHS

Table 3: Lengths of successful (best case) and failed (worst case) episodes in Mortar Mayhem. The values are calculated by equations (1-6).

|  | MM | MMAct | MMGrid | MMActGrid |
|---|---|---|---|---|
| Min (Lower Bound) | 38 | 19 | 46 | 7 |
| Max (Upper Bound) | 135 | 115 | 119 | 79 |

$$ClueTask = (ShowDuration + ShowDelay) \cdot CommandCount \qquad (1)$$
$$ActTask = (ExecutionDuration + ExecutionDelay) \cdot CommandCount \qquad (2)$$
$$ActTask = ActTask - ExecutionDelay + 1 \qquad (3)$$
$$MaxEpisodeLength = ClueTask + ActTask \qquad (4)$$
$$ActFailure = ExecutionDuration + 1 \qquad (5)$$
$$MinEpisodeLength = ClueTask + ActFailure \qquad (6)$$

Table 4: Lengths of successful episodes in Mystery Path, Mystery Path Grid, and Searing Spotlights. The values for MP and MPGrid are retrieved from one agent trained on the ground truth. The values for Searing Spotlights are based on an optimal policy. The optimal policy traverses the shortest path to the coin and then to the exit.

|  | MP | MPGrid | Searing Spotlights |
|---|---|---|---|
| Min (Lower Bound) | 23 | 6 | 15 |
| Mean | 39.13 | 13.17 | 35.33 |
| Std | 7.36 | 3.22 | 8.01 |
| Max | 73 | 29 | 67 |
| Upper Bound | 512 | 128 | 512 |
| Policy | Agent | Agent | Optimal |
| Episode Samples | 100k | 100k | 100k |

## C.3 RESET PARAMETERS TO CONFIGURE THE ENVIRONMENTS

Table 5: These are the default parameters that we used throughout this paper as default. Parameters with a * indicate that these are uniformly sampled. Values in square brackets are discrete choices, while values in parentheses consider a range to sample from. Mortar Mayhem Grid features the same parameters as its parent, but only the modified ones are presented.

| Mortar Mayhem | | Searing Spotlights | |
|---|---|---|---|
| **Parameter** | **Default** | **Parameter** | **Default** |
| Agent Scale | 0.25 | Max Episode Length | 512 |
| Agent Speed | 2.5 | Agent Scale | 0.25 |
| Arena Size | 5 | Agent Speed | 2.5 |
| Number of Available Commands | 9 | Agent Visible | False |
| Number of Commands* | [5] | Agent Health | 100 |
| Command Show Duration* | [3] | Sample Agent Position | True |
| Command Show Delay* | [1] | Use Exit | True |
| Execution Delay* | [6] | Exit Scale | 0.5 |
| Execution Duration* | [18] | Exit Visible | False |
| Hide Visual Feedback | False | Number of Coins* | [1] |
| Reward Command Failure | 0 | Number of Initial Spotlight Spawns | 4 |
| Reward Command Success | 0.1 | Number of Spotlight Spawns | 30 |
| Reward Episode Success | 0 | Spotlight Spawn Interval | 30 |
| **Mortar Mayhem Grid** | | Spotlight Spawn Decay | 0.95 |
| **Parameter** | **Default** | Spotlight Spawn Threshold | 10 |
| Number of Available Commands | 5 | Spotlight Radius* | (7.5-13.75) |
| Number of Commands* | [10] | Spotlight Speed* | (0.0025-0.0075) |
| Execution Delay* | [2] | Spotlight Damage | 1 |
| Execution Duration* | [6] | Light Dim Off Duration | 6 |
| **Mystery Path** | | Light Threshold | 255 |
| **Parameter** | **Default** | Hide Visual Feedback | False |
| Max Episode Length | 512 | Render Background Black | False |
| Agent Scale | 0.25 | Reward Inside Spotlight | 0 |
| Agent Speed | 2.5 | Reward Outside Spotlights | 0 |
| Cardinal Origin Choice* | [0, 1, 2, 3] | Reward Death | 0 |
| Hide Origin | False | Reward Exit | 1 |
| Hide Goal | True | Reward Coin | 0.25 |
| Hide Visual Feedback | False | Reward Max Steps | 0 |
| Reward Goal | 1 | | |
| Reward Fall Off | 0 | | |
| Reward Path Progress | 0 | | |
| Reward Step | 0 | | |

## C.4 MORTAR MAYHEM GRID TRAJECTORY

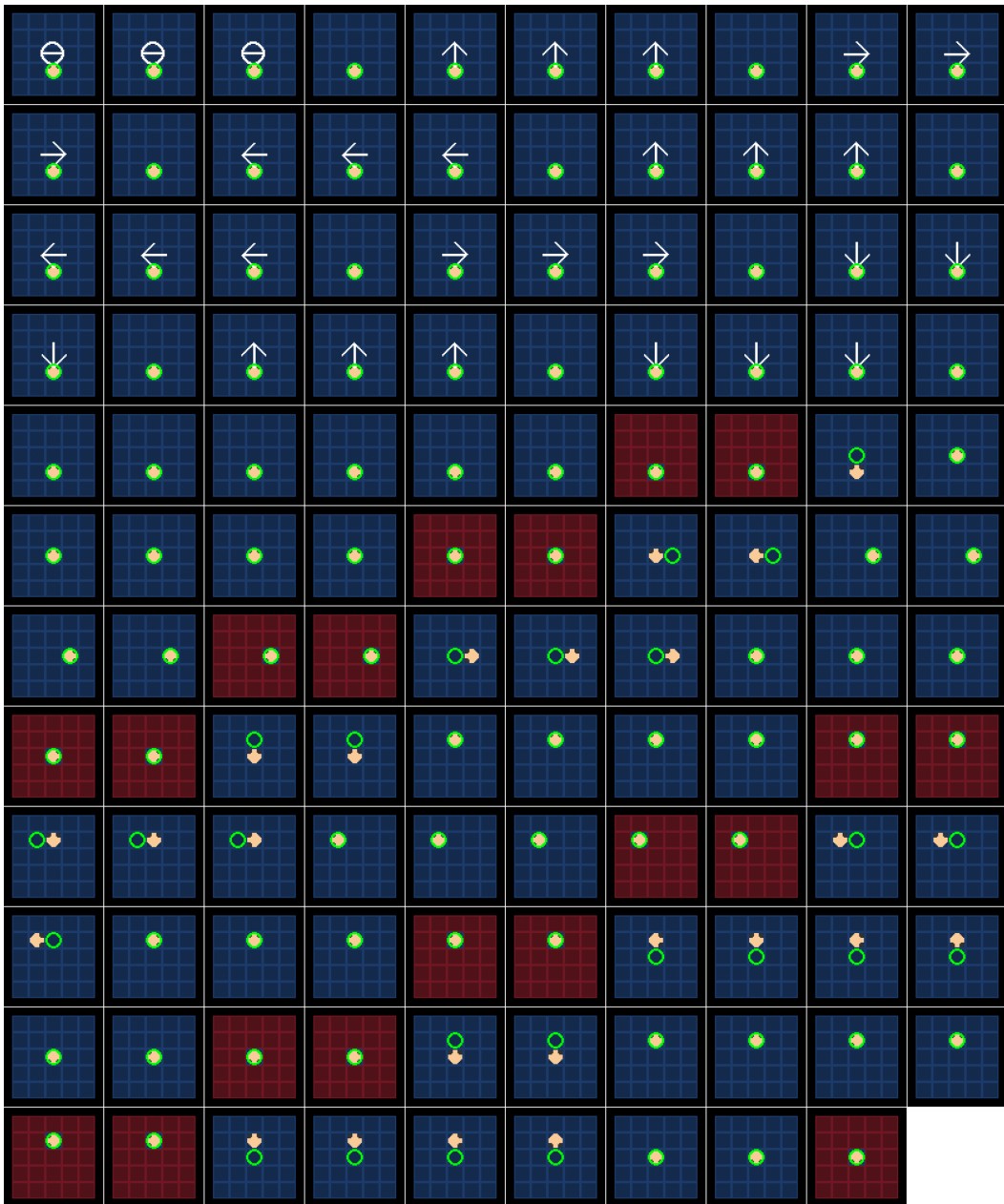

Figure 11: A successfully completed episode of Mortar Mayhem Grid showing the ground truth. The zip archive of the supplementary material provides an episode as a video.

## C.5 MYSTERY PATH GRID TRAJECTORY

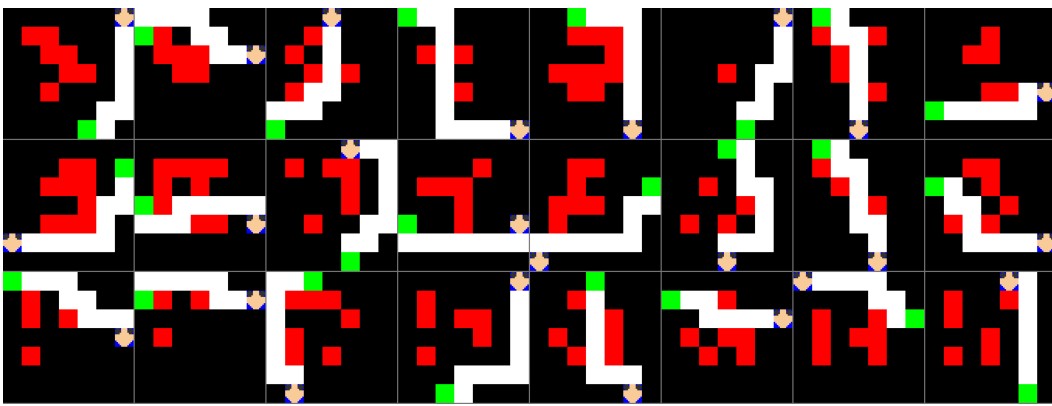

Figure 12: Various randomly generated levels in the Mystery Path environment.

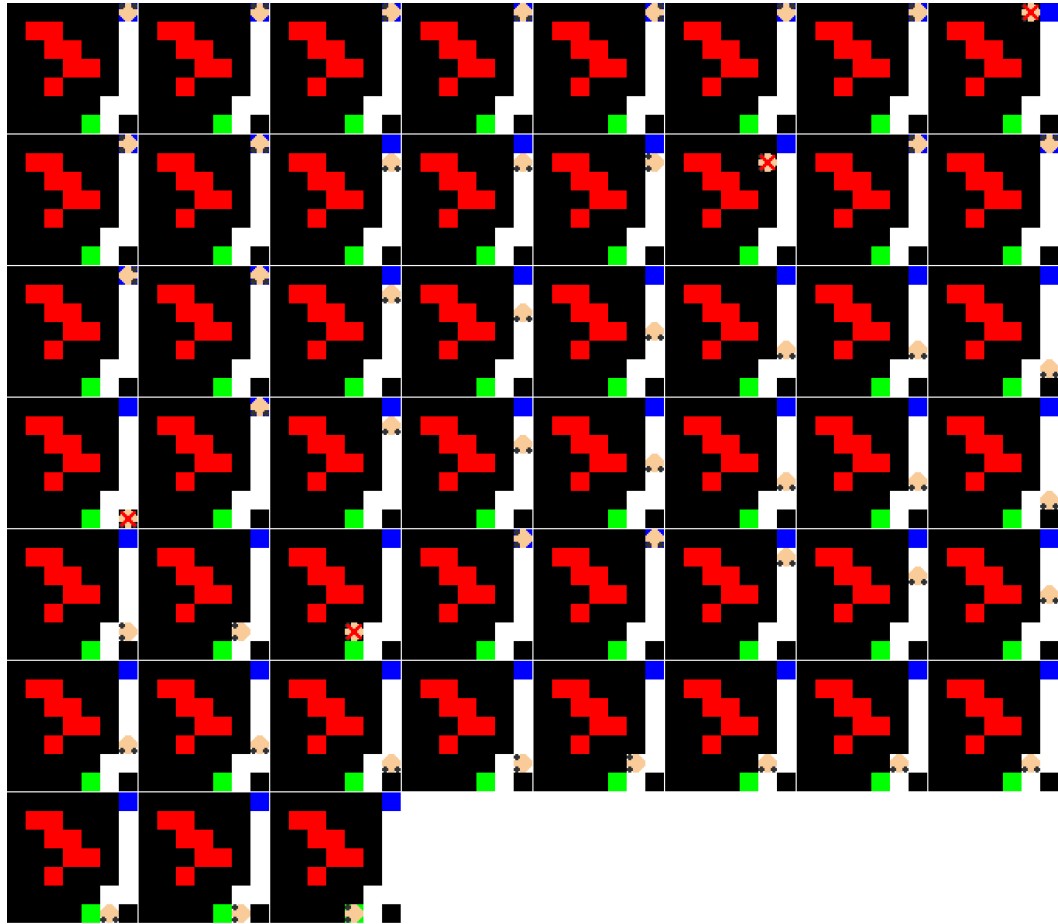

Figure 13: A successfully completed episode of Mystery Path Grid showing the ground truth. The zip archive of the supplementary material provides an episode as a video.

## C.6 SEARING SPOTLIGHTS TRAJECTORY

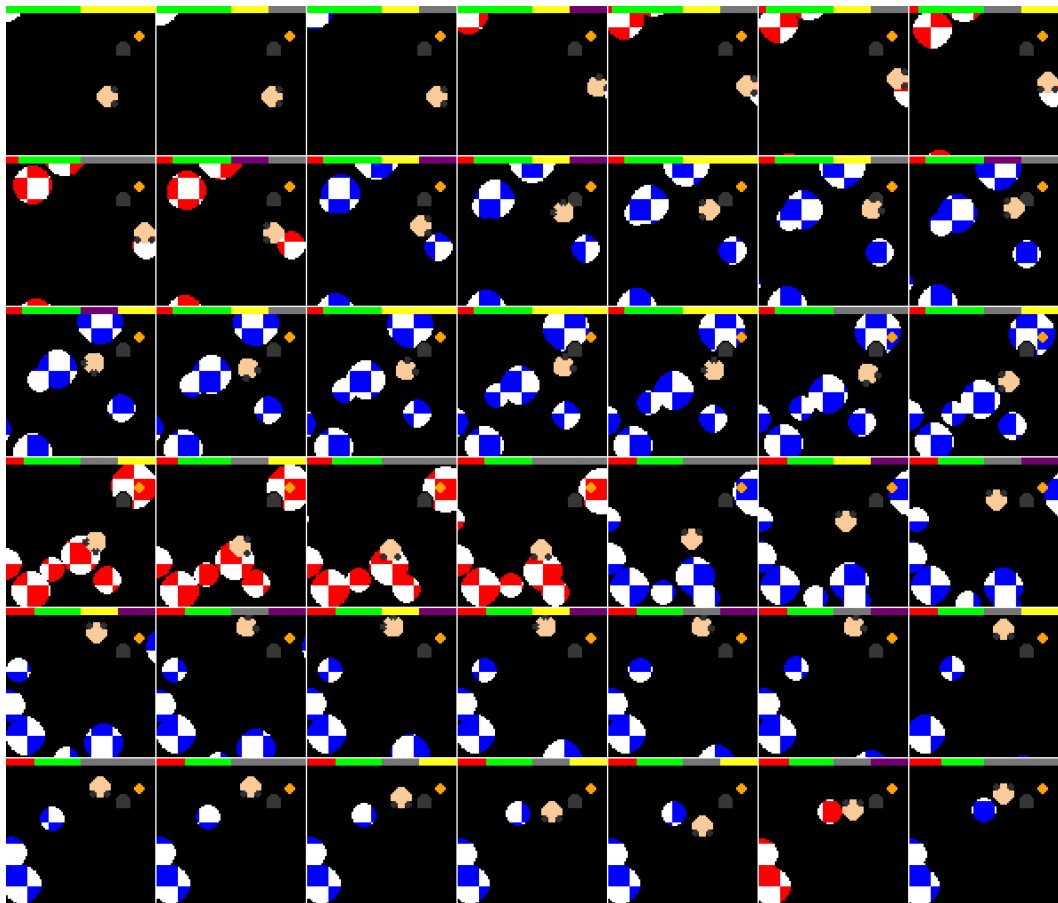

Figure 14: An excerpt of Searing Spotlights showing the ground truth and using a frame skip of four. The zip archive of the supplementary material provides an episode as a video.

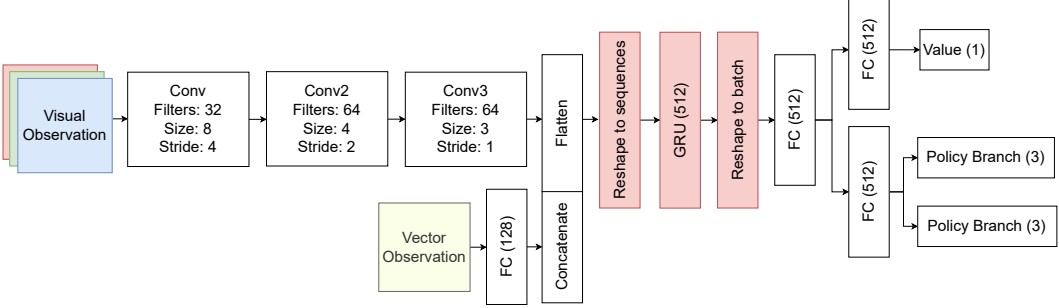

Figure 15: GRU-PPO utilizes an actor-critic feed-forward convolutional recurrent neural network. Visual and vector observations are encoded as an entire batch by either convolutional or fully connected layers. The encoded features are concatenated and then reshaped to sequences before feeding them to the recurrent layer. Its output has to be reshaped into the original batch shape. Further, the forward pass is divided into two streams relating to the value function and the policy. The number of policy heads is equal to the number of action dimensions given by a multi-discrete action space.

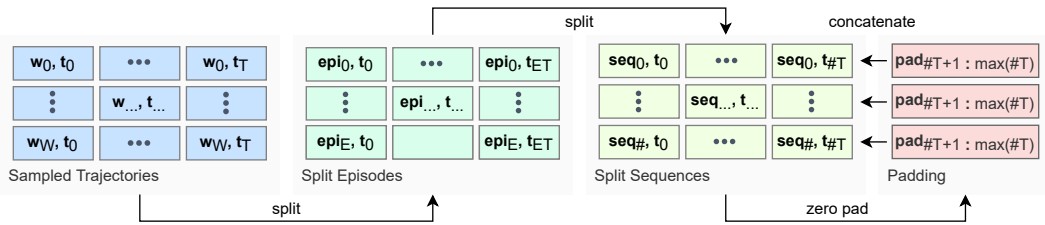

Figure 16: The data preprocessing starts out by sampling trajectories across $W$ workers for $T$ steps. Next, $E$ episodes of varying length $ET$ are extracted from the trajectories. Those can be further split into $\#$ sequences of varying length $\#T$. At last, zero padding is used to retrieve sequences of fixed length $max(\#T)$.

## D    BASELINES

### D.1    GRU-PPO

Our GRU-PPO baseline utilizes the clipped surrogate objective (Schulman et al., 2017). Due to the usage of the Gated Recurrent Unit as the recurrent layer, the to be selected action $a_t$ of the policy $\pi_\theta$ depends on the current observation $o_t$ and hidden state $h_t$ of the recurrent layer. $\hat{A}_t$ denotes advantage estimates based on generalized advantage estimation (GAE) (Schulman et al., 2016), $\theta$ the parameters of a neural net and $\epsilon$ the clip range.

$$L_t^C(\theta) = \hat{\mathbb{E}}_t[min(q_t(\theta)\hat{A}_t, clip(q_t(\theta), 1 - \epsilon, 1 + \epsilon)\hat{A}_t)] \tag{7}$$

$$\text{with ratio } q_t(\theta) = \frac{\pi_\theta(a_t|o_t, h_t)}{\pi_{\theta old}(a_t|o_t, h_t)}$$

The value function is optimized using the squared-error loss $L_t^V(\theta)$. $\mathcal{H}[\pi_\theta](o_t)$ denotes an entropy bonus encouraging exploration (Schulman et al., 2017). Both are weighted using the coefficients $c_1$ and $c_2$ and are added to $L_t^C$ to complete the loss:

$$L_t^{CVH}(\theta) = \hat{\mathbb{E}}_t[L_t^C(\theta) - c_1 L_t^V(\theta) + c_2\mathcal{H}[\pi_\theta](o_t, h_t)] \tag{8}$$

$$\text{with } L_t^V(\theta) = (V_\theta(o_t, h_t) - V_t^{targ})^2$$

IMPLEMENTATION DETAILS

Three major components have to be implemented: the forward pass of the model (Figure 15), the processing of the sampled training data (Figure 16), and the loss function (Equation 9).

The training data is sampled by a fixed number of workers for a fixed amount of steps and is stored as a tensor. Each collected trajectory may contain multiple episodes that might have been truncated. After sampling, the data has to be split into episodes. Optionally, these can be further split into smaller sequences of fixed length. Otherwise, the actual sequence length is equal to the length of the longest episode. Episodes or sequences that are shorter than the desired length are padded using zeros. As the data is structured into fragments of episodes, the hidden states of the recurrent layer have to be selected correctly. The output hidden state of the previous sequence is the input hidden state of its consecutive one. This approach is also known as truncated backpropagation through time (truncated bptt). Finally, minibatches sample multiple sequences from the just processed data.

Concerning the forward pass of the model, it is more efficient to feed every non-recurrent layer the entire batch of the data (i.e. $batch\ size = workers \times steps$) and not each sequence one by one. Whenever the batch is going to be fed to a recurrent layer during optimization, the batch has to be reshaped to the dimensions: number of sequences and sequence length. After passing the sequences to the recurrent layer, the data has to be reshaped again to the overall batch size. Note that the forward pass for sampling trajectories operates on a sequence length of one. In this case, the data keeps its shape throughout the entire forward pass.

Once the loss function is being computed, the padded values of the sequences have to be masked so that these do not affect the gradients. $L^{mask}$ is the average over all losses not affected by the paddings.

$$L^{mask}(\theta) = \frac{\sum_t^T \left[ mask_t \times L_t^{CVH}(\theta) \right]}{\sum_t^T [mask_t]} \tag{9}$$

$$\text{with } mask_t = \begin{cases} 0 & \text{where padding is used} \\ 1 & \text{where no padding is used} \end{cases}$$

## D.2 MEMORY-LESS PPO

In the case of memory-less PPO, Frame Stacking, and HELM, the recurrent layer is removed. The training data is not split into sequences. Loss masking is not used.

## D.3 FRAME STACKING

$n - 1$ past frames are stacked to the agents current visual observation. In the case of stacking 4 frames, RGB frames are used. Grayscale frames are utilized in the case of stacking 16 frames to reduce the model's input dimension.

## D.4 HELM

If HELM is used, the model receives another encoding branch that receives the current grayscale visual observation. The observation is then propagated by a frozen hopfield and a pre-trained transformer as described by Paischer et al. 2022. The resulting features are concatenated to the ones from the CNN and the vector observation encoder.

## D.5 CHOICE OF HYPERPARAMETERS

Table 6 enumerates the utilized hyperparameters. The agent in Searing Spotlights uses a frame skip of 2, which halves the length of an episode easing the agent's exploration. The quite large batch size is chosen to max out the wall-time efficiency of the utilized resources. One training based on GRU-PPO utilizes almost 32GB of VRAM. Most of the hyperparameters related to memory were found by running a grid search on an early version of Mortar Mayhem Grid. Besides that we made coarse hyperparameter sweeps especially for Searing Spotlights that did not yield any benefit. We

Table 6: These are the hyperparameters that we used for all training runs. The sequence length is dynamically set by the longest episode inside the batch of the gathered training data. The experiments on Searing Spotlights and Mystery Path utilized a fixed sequence length of 128. The learning rate and the entropy coefficient decay linearly.

| Hyperparameter | Value |
|---|---|
| Training Seeds | 100000 |
| Sequence Length | Max |
| Number of Workers | 32 |
| Worker Steps | 512 |
| Batch Size | 16384 |
| Number of Mini Batches | 8 |
| Mini Batch Size | 2048 |
| PPO Updates | 10000 |
| Training Steps | 163840000 |
| Discount Factor Gamma | 0.99 |
| GAE Lamda | 0.95 |
| Epochs | 3 |
| Value Loss Coefficient | 0.25 |
| Max Gradient Norm | 0.5 |
| Clip Range Epsilon | 0.2 |
| Initial Learning Rate | 3e-4 |
| Final Learning Rate | 1e-4 |
| Initial Entropy Coefficient | 1e-4 |
| Final Entropy Coefficient | 1e-5 |
| HELM Beta | 1000 |
| Optimizer | AdamW |

tried the values 50, 100, 200, and 1000 for HELM's beta hyperparameter on MMActGrid. All values ended up with a nearly identical performance. We ultimately chose 1000 because there is not much variance in individual observations in Mortar Mayhem and Mystery Path. Overall, we rather allocated our resources to scale the difficulty of Mortar Mayhem and Mystery Path, while investigating the issues of Searing Spotlights.

# E  WALL-TIME EFFICIENCY OF HELM

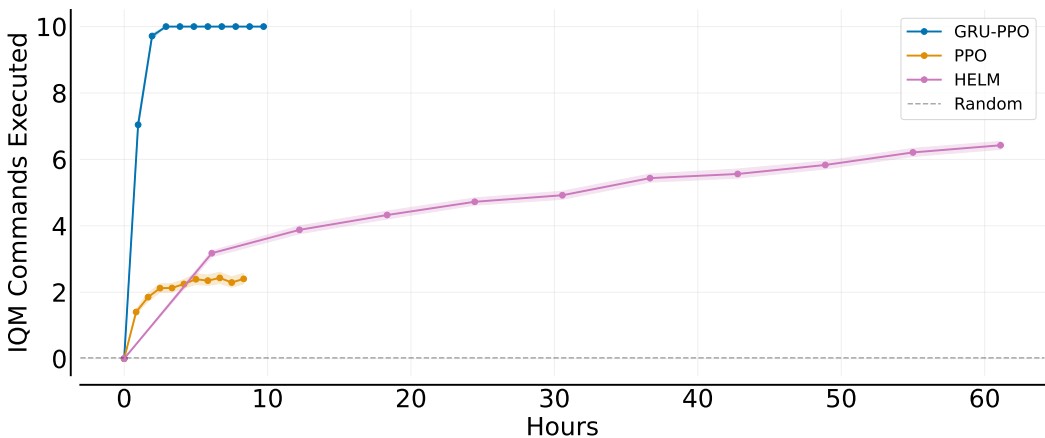

Figure 17: Generalization performances on novel seeds in Mortar Mayhem Act Grid showing the wall-time efficiency of PPO, GRU-PPO, and HELM. These experiments are run on an NVIDIA A100 Tensor-Core-GPU and an AMD EPYC 7542 CPU (32 cores).

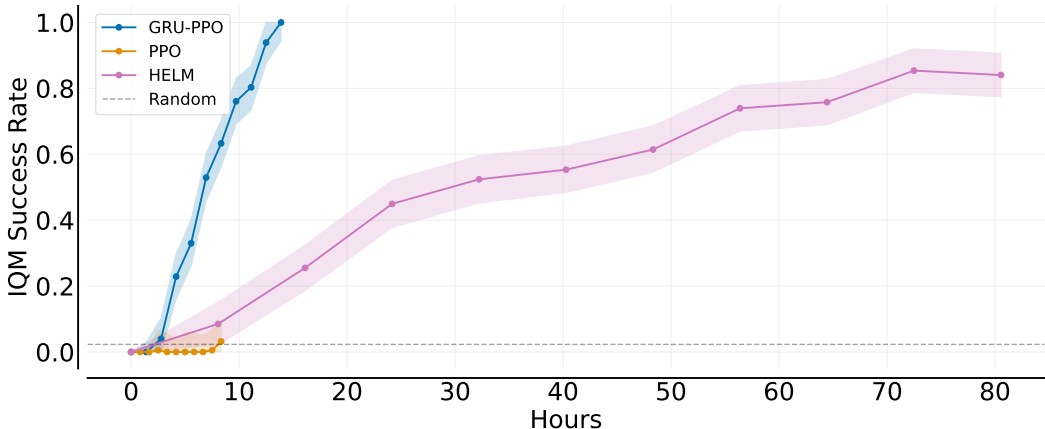

Figure 18: Generalization performances on novel seeds in Mystery Path Grid (hidden goal) showing the wall-time efficiency of PPO, GRU-PPO, and HELM. These experiments are run on an NVIDIA A100 Tensor-Core-GPU and an AMD EPYC 7542 CPU (32 cores).

# F  MEASURES TAKEN ON SEARING SPOTLIGHTS

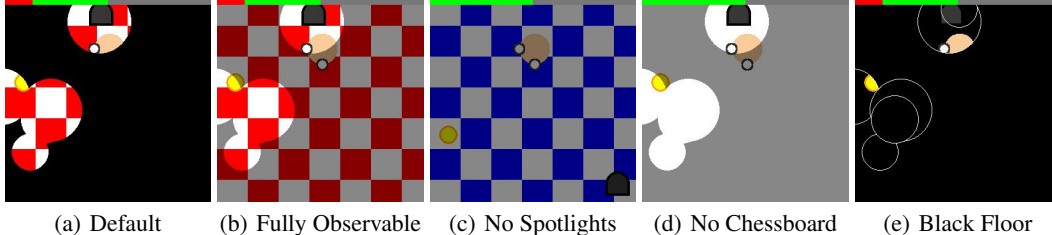

(a) Default  (b) Fully Observable  (c) No Spotlights  (d) No Chessboard  (e) Black Floor

Figure 19: Varying visual representations of Searing Spotlights are examined. (a) poses the original task. (b) provides perfect information, because the global light is just dimmed. (c) disables the spotlight dynamics while dimming the global light until off. (d) removes the tiled background. (e) renders the floor black that requires to render the spotlights' circumference white.

Multiple experiments were conducted to ablate the environment's difficulty and to vary the model architecture. Figure 19 shows a few variations of the visual representation of Searing Spotlights. Whenever spotlights are present, the recurrent agent fails. Subsequently, we enumerate the measures that we tried with limited success to help the agent to learn a more meaningful policy. Increasing the scale of all entities but the spotlights or reducing the number of spotlights may have undesirable consequences to the task's quality. Random policies may be more successful under these measures.

- **Reduce episode length to ease exploration**
    - Leverage frame skipping
    - Raise agent speed
    - Scale up all entities except spotlights
- **Vary spotlights**
    - Add negative reward of $-0.01$ for the agent being inside the spotlights
    - Stationary spotlights
    - Fewer spotlights
    - Constant spotlight size and speed
- **Vary task**
    - Use only one coin and no exit
    - Use the exit but no coin
    - Agent always starts at the center of the level
    - Increase agent health points
    - Add negative death reward of $-0.1$
- **Vary agent observation**
    - Environment is fully observable for few steps
    - Make the agent, coin, exit, or all of them permanently visible

- Add health points bar
- Render the agent's last action onto to the observation instead of feeding a one-hot encoded feature vector
- Feed the agent its exact position as normalized scalars
- Slightly dim the light (perfect information)
- Render the environment's floor black while drawing the spotlights' circumference white

- **Vary model architecture**
    - Add residual connection around the GRU cell
    - Use LSTM instead of GRU
    - Reduce sequence length from 128 to 64
    - Use Impala CNN Espeholt et al. (2018) instead of Nature CNN Mnih et al. (2015)
    - Add fully connected layer between CNN and GRU

## G    SOURCE CODE AVAILABILITY OF TRANSFORMER-BASED APPROACHES

During the rebuttal, few complex frameworks were brought to our attention that claim to support a DRL algorithm leveraging Gated TransformerXL (Parisotto et al., 2020) or just TransformerXL, namely Rllib, DI-engine, and Brain Agent. We tried to use Memory Gym with RLib but observed questionable outputs like negative rewards (even though the environments did not use any negative rewards). The framework is too complex to analyze and eventually to verify. The same accounts for Brain Agent, which is not densely commented nor documented. DI-Engine implements an R2D2 (Kapturowski et al., 2019) variant based on Gated TransformerXL. We tried it and learned that it has a poor sample throughput by a magnitude of 10 if compared to GRU-PPO. We do not claim that those frameworks are dysfunctional, because we just report our limited experience by exploring those.

By the time of acceptance, we finally implemented a lightweight and easy-to-follow TransformerXL + PPO baseline, which is now used in our momentary work on Memory Gym. Source Code: `https://github.com/MarcoMeter/episodic-transformer-memory-ppo`

