# OpenReview forum: "Memory Gym: Partially Observable Challenges to Memory-Based Agents"
_ICLR.cc/2023/Conference — ICLR 2023 poster_

### Official Review · Reviewer_aYs5 · 2022-10-18

**Confidence:** 3
**Correctness:** 3
**Technical Novelty And Significance:** 1
**Empirical Novelty And Significance:** 3
**Recommendation:** 5

**Clarity, Quality, Novelty And Reproducibility:**

- The quality of the work is good (especially the code implementation). The manuscript is clear in making a case for this benchmark

- The work being a benchmark, it does not offer technical novelty. Its usefulness for the community is unclear or at least not clearly demonstrated given the current state of the work (see discussion on memory, sequence lengths and RNNs above).

- The manuscript lacks clarity in presentation and depth in defining concepts that are being manipulated. For example, all performance metrics presented (fig 4,5,6c,15,16) are IQM (commands, success rates, returns) which are used without any definition (the acronym itself appears without any prior presentation).

- The videos, provided in batch form without any explanation, do not help the reviewer to appreciate the intricacies of the work or acquire a clearer understanding of what is of interest in what is being shown.



**Strength And Weaknesses:**

Strengths:
The work tackles a very important problem that is of great interest to the Recurrent Neural Network (RNN) community, which is long term dependency learning, and large sequence memorisation, in the context of memory requiring Deep Reinforcement Learning (DRL) tasks.
The authors put notable effort into ensuring their environments require sequence memorisation capabilities to be successfully completed. They also clearly state a case for the need for such a benchmark among the plethora of available environments to specifically tackle this issue. Indeed:
- Code is well written and simple enough to use.
- The desiderata is met, especially in terms of enforcing a dependency between success and long term memory decision making.
- Difficulty scaling for the games enables benchmarking memory capabilities of simplistic models as well as larger architectures.

Weaknesses:
The paper presenting a new benchmark, it is naturally void of any technical novelty. However, it is unfortunate that it also lacks technical precision for many of the concepts manipulated:
- The word memory itself, which gives the benchmark its whole purpose, is used 123 times throughout the paper without being clearly defined, or more specifically, put into context. It is evident that the authors refer to maintaining a representation of information from long sequence of past grids and its use for future decisions, however one has to refer to Fig 14 in appendix D on page 20 of the manuscript for the technical representation of the recurrent neural network block in the DRL architechture.
- Thus, a clearer presentation of memory mechanisms in RNNs and their relationship to DRL algorithms would be much welcome to elucidate what exactly is being evaluated by the benchmark. With a plethora of  advances on RNN memory capabilities for sequence modelling (see works such as "Efficiently Modeling Long Sequences with Structured State Spaces" by Gu et al, and "Liquid Structural State-Space Models" by Hasani, Lechner et al), any performance evaluation should be based on a quantitative measuring of the dimension and sequence lengths of information required to achieve the task against the memorisation capabilities of the RNN network used in the PPO architecture (or any other DRL method relying on RNNs for memory).
- With memory capabilities of state-of-the-art RNNs allowing successful learning of Long Range Arena tasks such as Path-X (16384 sequence length), it would be interesting to have a more extensive quantitative discussion about how increasing difficulty levels of the games affects the dimensions of the sequences that need to be memorised. This would justify why this benchmark can be useful in the future as we develop increasingly memory capable architectures.
- Though not at the core of the work, the authors only test GRU and LSTM architectures. Future directions evoke implementing more elaborate models. However, it would have been interesting to see better adapted architectures used especially for the Searing Spotlights environment.

**Summary Of The Paper:**

The authors propose a benchmark to solely focus on benchmarking Deep Reinforcement Learning agents' memory and ability to generalise. In addition to open source accessibility, simulation step speed and flexibility (level hardness, noise etc) the authors claim their benchmark is better suited to distinctly evaluate the role of memory in DRL performance. They show the need for RNN implementations of PPO taking learning from sequences to successfully complete (2/3) of the games where memory-less implementations fail.

**Summary Of The Review:**

The work is of high relevance for benchmarking a very important aspect of learning which is the capability of networks and in this case reinforcement learning agents to achieve tasks requiring long sequence memorisation. The manuscript itself is generally well written, but lacks some clarity and rigour in manipulating core ideas to the case it is making. With additional elucidation of links between the proposed tasks and the quantitative increase in memorisation required for solving them, this work could prove to be suitable for benchmarking models not only today, but into the future as models improve. This potential is still vague in the current state of the work.

---

> ### Author Response · Authors · 2022-11-11
> **Follow-up questions conerning potential revisions**
>
> We enjoyed reading and discussing your review, thank you! It provides a refreshing perspective on our work as well as valuable feedback for improvement. We currently prepare the revision and run further experiments. To ensure the best possible revision, we've got a few questions.
>
> > The word memory itself [...] is used [...] without being clearly defined [...]
>
> Are you looking for extensive reasoning, or is it sufficient to further narrow the definition in the introduction to avoid misunderstandings?
>
> > [...] clearer presentation of memory mechanisms in RNNs and their relationship to DRL algorithms would be much welcome [...]
>
> We can add formal, algorithmic, and implementation details of our GRU-PPO baseline to the annex. Would this possibly resolve your concern? We did not allocate much space for this baseline in the main body because we intend to focus on the proposed environments and their traits.
>
> > [...] any performance evaluation should be based on a quantitative measuring of the dimension and sequence lengths [...].
>
> This comment is inspiring and you are right. We will revise our paper to make the dimensions of those sequence lengths apparent. Reinforcement Learning is a pretty dynamic setting. Depending on the environment, episodes can be small or long. Thus, the sequence length may vary based on the policy's performance. In Mortar Mayhem, the sequence gets longer as the agent's policy improves (until an upper limit is reached based on the environment's configuration). Mystery Path and Searing Spotlights are the opposite in this context. Successful policies will lead to shorter episodes, while unsuccessful ones will cause episodes to be pruned by a time limit.
>
> > However, it would have been interesting to see better adapted architectures used especially for the Searing Spotlights environment.
>
> Would you mind sharing what architectures you are referring to?
>
>
> Thanks!

---

> > ### Comment · Reviewer_aYs5 · 2022-11-17
> > **Reply**
> >
> > Thank you for your return,
> >
> > In terms of defining memory, I think narrowing down the definition in the introduction is fine as long as a connection is made to the third point mentioned about the dimension and sequence lengths of the data that is needed for successful planning.
> >
> > Now for the second and fourth points, formal, algorithmic, and implementation details would be very welcome in the annex along with a discussion on other potential model types.
> > In addition to the LSTM and GRU-ODE architectures, authors could plug in other RNN models such as the ODE-RNN (continuous time vanilla RNN by Y. Rubanova et al 2019), or Closed-form Continuous-time Neural Models (Hasani et al 2021).
> > Also, the authors could perhaps tweak the architecture to accommodate models with state of the art long sequence modelling architectures such as the S4 model (Gu et al 2021) or newer variants such as Liquid-S4 (Hasani & Lechner et al 2022) which offer unprecedented memorisation scale capabilities.

---

### Official Review · Reviewer_guu9 · 2022-10-20

**Confidence:** 4
**Correctness:** 4
**Technical Novelty And Significance:** 3
**Empirical Novelty And Significance:** 3
**Recommendation:** 8

**Clarity, Quality, Novelty And Reproducibility:**

The writing and presentation is extremely clear, kudos.

The quality seems overall high.

The novelty of the benchmark is moderate given the existing work, but I think the authors do a decent job of describing the unique value of their benchmark relative to existing tasks. And the results with the searing spotlights are interesting scientifically, which boost the value a bit.


**Strength And Weaknesses:**

Strengths:
* Presentation is very clear.
* Tasks are interesting and deliver distinct  memory challenges.
* Scalable difficulty is a very nice attribute for future research.
* Results with spotlights + recurrence are fascinating.

Weaknesses:
* If possible, it would be nice to include more baselines from different algorithmic families, such as recurrent IMPALA or VMPO (used in the Fortunato et al. and Parisotto et al. papers, IIRC—of course the latter uses transformers, but VMPO does not require them).


**Summary Of The Paper:**

This paper outlines some desiderata for memory tasks for RL, and proposes/releases a set of three varied benchmark tasks that meet these desiderata. The paper runs a variety of baselines on the tasks and shows that they have nice properties such as strong memory dependence and scalable difficulty. One task, searing spotlights, leads to results that are particularly interesting from a scientific perspective.

**Summary Of The Review:**

I think this paper offers an interesting new benchmark for memory tasks, and some independently interesting results. The authors do a decent job of motivating their work relative to prior work. There could always be more baselines, etc. but overall I find the paper to be a useful contribution.

---

> ### Author Response · Authors · 2022-11-11
> **Response**
>
> Thanks for your review and positive sentiment towards our contributions.
>
> > If possible, it would be nice to include more baselines from different algorithmic families, such as recurrent IMPALA or VMPO [...]
>
> Unfortunately, we cannot run further algorithms during the limited time of the revision. Even if we found a plug-and-play implementation, we  would still need a suitable search of hyperparameters. Just out of curiosity, do you know of anybody who was able to reproduce V-MPO (Song et al. 2020)? So far we were unable to find an implementation that can reproduce its originally reported results.

---

> > ### Comment · Reviewer_guu9 · 2022-11-17
> > **VMPO**
> >
> > Good point, I asked around a bit from people I thought might have used it, and nobody knows of an open source reproduction. Very reasonable to not compare under those circumstances.

---

### Official Review · Reviewer_Arip · 2022-10-24

**Confidence:** 4
**Correctness:** 4
**Technical Novelty And Significance:** 2
**Empirical Novelty And Significance:** Not applicable
**Recommendation:** 6

**Clarity, Quality, Novelty And Reproducibility:**

The paper is basically written well, and the ideas are clearly addressed. There is no new algorithm/model proposed, but a new task set for POMDP studies. The code will be open source.



**Strength And Weaknesses:**

**Strength**
- The being addressed problem is important, that is, the lack of a standard benchmark for POMDP
- The proposed environments have good properties (Table 1)
- The proposed tasks are interpretable and can be customized according to the research demand
- The proposed tasks can test model's capacity of generalization
- The authors conducted experiments to support their claims


**Limitations**
- The task scope is limited to 3 Atari-like games with 2D vision as input. There is no environment with continuous action space. However, due to the large diversity of POMDP research, e.g., robot control and multi-modality observations, the proposed framework still has not solved the lack of a standard POMDP benchmark.
- The experiments can be improved. Some of possibly useful trials are as follows: (1) Testing feedforward NN with $N$-steps observations as input. The result of how performance changes with $N$ can provide clues of how much long-term memory (vs. short-term memory) is required. (2) Testing some off-policy and probably more powerful algorithm of memory-dependent RL such as R2D2 (https://openreview.net/forum?id=r1lyTjAqYX) and ACER (https://arxiv.org/abs/1611.01224v2).







**Summary Of The Paper:**

As a standard benchmark for POMDP tasks is desired due to rising research interests, the paper proposed 3 new Gym tasks which are memory-dependent POMDPs. The key features of the environments are (1) "strongly depend on memory".  (2) different levels of difficulty. (3) good meta properties (Table 1).

The authors conducted baseline experiments using PPO as RL algorithm and GRU for history extraction and compared with PPO + feedforward NN and HELM. It can be seen that the tasks cannot be solved without memory. The results also show various difficulty levels of the tasks even using PPO+GRU. Therefore, the authors conclude that the proposed tasks yield a new benchmark for memory-dependent POMDP research.








**Summary Of The Review:**

The paper proposed a new task set for memory-dependent POMDP studies. The tasks have many desired properties, and the experiments proved that memory is necessary to solve them. Meanwhile, the task scope is limited, and the experiments could be more comprehensive and convincible by testing more algorithms and models. In sum, my recommendation is a borderline reject.

-------- Updated after rebuttal -------

See my comments below

---

> ### Author Response · Authors · 2022-11-13
> **Response**
>
> Thank you for your valuable feedback and comments!
>
> ### N-steps experiments
>
> We are currently running experiments using frame stacks in the range of $4$ to $16$ frames, which will rather cover short-term memory aspects of the environments. Larger frame stacks do not seem intuitive for the basic feed forward model architecture of PPO. This is more likely to be addressed when using transformer architectures of fixed sequence lengths. However, transformer-based baselines are left to future work.
>
> ### Standard POMDP benchmark
>
> A large and diverse POMDP benchmark suite has quite a huge scope. Certainly, multi-modality observations and robot control tasks are important research directions. Our focus is limited to assess the agent's memory under sequential decision making. For this narrow goal, we do not examine different action spaces. We could add continuous action spaces to our environments to allow people to just plug in their agents without further ado (e.g. discretizing actions, implementing discrete action spaces). This would be rather a comfort feature to make Memory Gym's environments more accessible. However, we consider continuous action spaces to result in very different problems that are not easy to compare with discrete spaces problems and thus shall be addressed explicitly in its own right, not as appendix of a discrete environment.

---

> > ### Comment · Reviewer_Arip · 2022-11-19
> > **Thanks**
> >
> > Thanks for the response. I am waiting for the N-steps experiment results.

---

> > > ### Author Response · Authors · 2022-11-19
> > > **Frame stacking results are available**
> > >
> > > Our revision should be available since yesterday. Figure 4 contains the requested frame stacking baseline results.

---

> > > > ### Comment · Reviewer_Arip · 2022-11-20
> > > > **Thanks**
> > > >
> > > > Thanks for the reminder and the N-steps results. I would like to enhance my rating. Meanwhile, although I agree with the authors that *continuous action spaces to result in very different problems that are not easy to compare with discrete spaces problems*, the narrow scope of the current environments cannot qualify the paper for a score of 8.  Thus, my updated recommendation is borderline accept.

---

### Official Review · Reviewer_e7qG · 2022-10-24

**Confidence:** 4
**Correctness:** 2
**Technical Novelty And Significance:** 2
**Empirical Novelty And Significance:** Not applicable
**Recommendation:** 3

**Clarity, Quality, Novelty And Reproducibility:**

### Major Points

- Proper baseline comparison:
  Framestacking is a simple baseline that should be included to highlight long-term memory dependence of the environments.
  This ensures no other short-term aspects of the environment can be exploited and strengthens the claim for long-term dependencies.
  Codebases of other state-of-the-art approaches are actually publicly available:

  Several open-source implementations of GTrXL are available:
  https://github.com/opendilab/DI-engine
  https://github.com/dhruvramani/Transformers-RL

  Codebase for HCAM:
  https://github.com/deepmind/deepmind-research/tree/master/hierarchical_transformer_memory

- Unclear presentation of results:
  The presentation of results conflates generalization and memory capability, making it difficult to interpret results.
  It would be much easier to follow if those two effects would be disentangled, i.e. evaluate memory capabilities on the entire level distribution and generalization on a train/test split of seeds.
  Further, the number of training seeds are not mentioned in the main paper, however this information is crucial to interpreting generalization results and the apparent gap.
  A helpful analysis here would be to show the generalization gap over different number of training seeds as in [2].
  Also, why are only 10 novel seeds chosen for evaluation, but repeated 5 times?
  It would be much more intuitive to evaluate on a higher number of seeds to get a better performance estimate.
  The variance of the Random baseline should be included in all plots.
  Is Figure 4c actually correct?
  How can it be that PPO reaches performance on-par with a random policy, but generalizes on novel seeds?
  If the plot is actually correct it raises the question whether this environment is really dependent on memory, since PPO is the best performing method.
  The authors mention performance of GRU-PPO and PPO for their environment ablation in Fig 6a, but do not show any learning curves.
  These learning curves would be much more important for interpreting results than the performance on Fig 6b.

- Interpretation of results:
  There is a huge generalization gap for PPO on MPGrid, while it solves the task in 50% of the cases during training.
  Since the goal is always placed on the opposing side of the origin and the solution is mostly the shortest path then this gap from training to test seeds is difficult to explain.
  Why does the performance of GRU-PPO stagnate when going from a per-tile to continuous movement of the agent?
  A possible reason for that is limited exploration, and could be alleviated by running a proper hyperparameter search for each method on each environment.
  Adding additional rewards, as in MPDense, alters the optimization problem and the optimal policy may change.
  If the immediate reward overshadows the reward of reaching the goal tile, the agent may learn to exploit the immediate reward only and will not perform well on novel seeds.
  Since GRU-PPO has shown performance lower than random on Searing Spotlights, it did not learn anything meaningful.
  Not only is GRU-PPO incapable of solving Searing Spotlights, but all baseline methods are.
  The authors identify the reason for that as vulnerability of GRU-PPO to spotlight perturbations, but another reason could simply be suboptimal hyperparameters.
  If recurrence is the sole reason, HELM should capable of solving the environment, right?

- Computational resources were allocated to scale the hardness of the environments:
  Baselines should be tuned on the respective environments before scaling the hardness, otherwise it is not clear whether bad performance is caused by inappropriate hyperparameters.

### Minor Points:

- The authors claim that "if an environment is solvable to some extent using an agent without memory, it is not easy to differentiate whether the memory mechanism is working"
  This is not true, it shows that a Markovian policy can exploit structure in the environment without relying on memory.
  Whether a memory mechanism is working is ill-defined here, how is it defined that a memory mechanism "works"?
  If a recurrent agent significantly outperforms a markovian one, it indicates that the addition of memory alters the optimization problem such that the task is easier to solve.

- The authors claim that DM Alchemy, Crafter, or ObstacleTower require non-trivial additional components.
  What are those?
  Why can't GRU-PPO simply be applied to those?

- Why is open-sourceness explicitly stated as requirement?
  Is it not sufficient to access a benchmark via an API without access to the source code?
  Obviously open-sourceness should be a requirement for published algorithms and novel methods, but is it required for benchmarks as well?
  Further, why is headless a requirement?
  Why would HPC facilities not support dependencies such as xvfb, or EGL?

- How can an environment be turned into a supervised learning problem?

- What does it mean that an "agent is left in uncertainty"?

- The authors write "A single observation leaves the agent clueless about which of the nine tiles to move next" in MM.
  This is not true, in fact, the last action can be inferred by the direction the agent is facing (the eyes in Fig 1), also the performance of PPO in Fig 4a indicates otherwise.
  PPO appears to be capable of exploiting this information to reach a score > 2 during training.

- What are differences between pits and walls in Mystery Path?

- The authors claim that mean cumulative reward is not an appropriate measure for task completion, why?
  A task is defined by its reward function and maximum reward corresponds to completion of a task.
  In the miner example the task is defined by collecting all diamonds and navigating to the exit afterwards, thus, if maximum reward is not achieved the level is not completed.
  Another claim: "Heist and Maze can be of strong dependence on memory, but only if tiny levels, which memory-less agents trivially solve, are excluded"
  Why does the appearance of easier levels in the level distribution alleviate the need for memory on harder ones?

- What input information do the different baselines receive? only observations or actions as well? what is the vector observation for the different environments?

- Has HELM been properly tuned? The appendix only mentions tuning of the beta parameter with no influence on performance. What about other hyperparameters such as learning rate?

- Phrases such as "MMAct needs twice as long to solve only five commands" should be rephrased to "GRU-PPO needs twice as long...", environments do neither converge nor solve a task.

- How is the "uncertainty of the environment" defined? Is it measured somewhere?

- Why are training and test seeds for MM shown in one plot in Fig 5a, but in separate plots for MP in Fig 5b and 5c?

- The authors claim that MiniGrid-Memory [3], Spot the Difference [4], or DM Ballet [1] are not well suited for evaluating memory.
  In the next sentence, they confirm that all of these environments require an agent to memorize cues in the very beginning of an episode, thus they can be used to evaluate the memory capability of an agent.
 The authors follow up on that by "Once this cue is memorized there is no need to manipulate the agent's memory further." and that it is sufficient to match extracted features to goal cues.
 In fact, most of the proposed environments in Memory-Gym don't require an agent to "manipulate" its memory, but rather match the features of the current observation with the features it has extracted during observing the sequence of actions in Mortar Mayhem, or the position of coin and exit in Searing Spotlights.
 It is not even clearly defined what "manipulating" an agent's memory means.


 [1] Andrew Kyle Lampinen, Stephanie C. Y. Chan, Andrea Banino, and Felix Hill. Towards mental time travel: a hierarchical memory for reinforcement learning agents, NeurIPS 2021.
 [2] Karl Cobbe, Christopher Hesse, Jacob Hilton, John Schulman, Leveraging Procedural Generation to Benchmark Reinforcement Learning, ICML 2020.
 [3] Maxime Chevalier-Boisvert, Lucas Willems, and Suman Pal. Minimalistic gridworld environment for openai gym, Github 2018.
 [4] Meire Fortunato, Melissa Tan, Ryan Faulkner, Steven Hansen, Adria Puigdomenech Badia, Gavin Buttimore, Charles Deck, Joel Z Leibo, and Charles Blundell. Generalization of reinforcemnt learners with working and episodic memory, NeurIPS 2019.

**Strength And Weaknesses:**

### Strengths:

- Potential relevance to the field:
  As the authors correctly point out, there is a variety of methods available, but no go-to benchmark to properly compare memory-based agents.
  Moreover, the memory dependence of some existing memory benchmarks is questionable.
- Reproducibility:
  Codebase is made publicly available.

### Weaknesses:

- Novelty:
  Mortar Mayhem is almost equivalent to the publicly available DM Ballet environment [1].
  Other existing benchmarks fulfill the same requirements of their desiderata as Memory-Gym, sometimes even with higher FPS, but no justification is given on why Memory-Gym should be the preferred choice.
  The memory dependency of some environments is questionable, i.e. PPO can exploit visual artifacts in Mortar Mayhem and Mystery Path, while being the best method on Searing Spotlights.
- Relevance of proposed benchmark:
  There are other environments that fulfill every point on the proposed desiderata, but it is not clear what advantage Memory-Gym has over them.
  Two of the three proposed environments are solved already within a moderate amount of training steps with a recurrent agent.
  The memory dependence of Mystery Path is questionable, since it can be solved in ~50% of the cases by a markovian policy as shown in Fig4b).
  The difficulty of Searing Spotlights stems from visual artifacts (random spotlights) rather than required memory, rendering it unsuitable for evaluating memory capabilities of an agent as proposed.
  Furthermore, difficulty scaling is achieved either through increasing required memory capacity (longer sequence of commands in MM, scaling grid in MP, etc.), or by increasing the complexity of the action space (continuous instead of grid-like movement).
  However, the authors state explicitly that evaluating the memory capacity can be done more effectively by other disciplines such as supervised learning, which contradicts their scaling approaches.
  Varying the time intervals between commands, in MM, however, seems to be a valid approach to scale the hardness with respect to what kind of information to store, instead of probing memory capacity.
- Presentation:
  The presentation of the environments is somewhat confusing and often lacks some context.
  It is not clear why some environment variants have different action spaces, diagonal movements would still be possible with the multi-discrete action space.
  Fig. 1 is very confusing: Why is the "Next Target" in Fig 1a) to the left while the command points to the right? Shouldn't the command indicate the direction towards the next target?
  It is not clear how the number of steps in Mortar Mayhem are computed.
  It is not clear how the movement of the agent is defined in the non-grid versions.
  Some unintuitive design choices were made without justification, e.g. negative feedback is only given visually instead of negative reward.
  From visual feedback the agent could never learn that e.g. Spotlights are actually bad as there is no negative reward for losing health.

**Summary Of The Paper:**

The authors point out drawbacks of current benchmarks that claim to evaluate memory in RL.
They propose a novel benchmark suite for evaluating the memory component of agents trained in partially observable environments.
The benchmark consists of three novel environments that vary in their difficulty and can be scaled to much harder difficulty.
Baseline results indicate that two of the three environments are solvable by a recurrent agent, while the third is not solvable.
The authors identify the reason for that as vulnerability of recurrence to random perturbations that are part of the environment.

**Summary Of The Review:**

The paper addresses an important issue in the field of partial observability in RL, namely to introduce a benchmark for evaluating memory capabilities of novel and existing methods.
Despite a well-structured desiderata, the authors fail to clarify the benefit of their proposed benchmark over existing ones.
Further, the design choices of the different environments in the benchmark are very confusing and not always justified or even contradictive.
The provided baseline results are difficult to interpret and not very convincing, making it difficult for new methods to reference these results.
They also indicate that some environments are exploitable by a markovian policy to a certain degree.
Finally, both environments and baseline results are presented poorly and would require a major revision in order to make the proposed benchmark usable by the RL community.

---

> ### Author Response · Authors · 2022-11-15
> **Response #1**
>
>
> Thank you for you very detailed comments to our work! These are helpful to establish our revision.
>
> ### Memory Gym versus Related Benchmarks
> > Other existing benchmarks fulfill the same requirements of their desiderata as Memory-Gym, sometimes even with higher FPS [...].
>
> > [...] the authors fail to clarify the benefit of their proposed benchmark over existing ones.
>
> Which related environments are you referring to that we did not cover in our work? This information would be helpful  to respond to your concerns and especially to be able to revise our work. We are happy to further improve our work.
>
>
> ### Equivalence between Mortar Mayhem and DM Ballet
> Mortar Mayhem’s first task is similar to DM Ballet: the agent is unable to move and has to memorize visual cues. The major difference lies in Mortar Mayhem’s second task. The second task in DM Ballet is short (2 steps at minimum and 4 steps at maximum needed) and does not require the agent to store more information in its memory. Mortar Mayhem’s Act Task cannot be solved by a markovian policy, while the episode lasts much longer. DM Ballet could be made more similar by modifying its second task. Instead of moving to the asked dancer, the agent could be asked to perform the same choreography.
>
> ### Questionable memory dependency in Mystery Path Grid
> A result of approximately $50\%$ in Mystery Path Grid by a markovian policy under training seeds is not satisfying at all. Lets consider a T-Maze (e.g. MiniGrid-Memory) with a good and a bad exit. A markovian policy finding the correct exit by chance is not satisfying as well. Though, we expect drops in performance if the markovian agent has to deal with a lower time budget.
>
> ### Exploiting Visual Artifacts
> In general, if agents have influence over their next observation, they can exploit it as some kind of memory. The observation could be considered as memory and the agent's action space as tool to write to this kind of memory. In such cases the agent might leave itself cues to utilize during its next decision. This effect may be enlarged by stacking past frames or adding the agent's last reward or last action to the observation space. We do not consider such abilities as bad. This potential is great for such policies.
>
> When considering Memory Gym's grid environments, the agent cannot infer its past action from its rotation. Based on its momentary observation, it cannot determine whether it rotated left, rotated right, moved forward, or did nothing at all. In the case of the multi-discrete action space and if the agent did not choose the no-op action, the agent can mostly infer its past action from its rotation. Searing Spotlights is the only environment that utilizes the agent's past action as part of its observation space. Once in the dark, the agent has to approximate its current position by recalling to a past visible position and its past actions. In general, exploiting the agent's last action may yield short-term benefits. Again, this is not the case for the grid environments that provide shorter episodes.
>
> ### Publicly Available Baseline Implementations
>
> The repositories [https://github.com/dhruvramani/Transformers-RL](https://github.com/dhruvramani/Transformers-RL) and [https://github.com/deepmind/deepmind-research/tree/master/hierarchical_transformer_memory](https://github.com/deepmind/deepmind-research/tree/master/hierarchical_transformer_memory) are incomplete. Vital algorithmic details are missing that might have been missed by the original publication. Such model architectures are not plug-and-play. It is quite an effort to achieve a flawless implementation because of the lack of algorithmic details. Reproducing GTrXL results would also take into account reproducing V-MPO. To the best of our knowledge nobody was able to reproduce V-MPO. We were unaware of DI engine. Thanks for sharing it! We will analyze its utility to potentially inspire our consecutive work.

---

> > ### Author Response · Authors · 2022-11-15
> > **Response #2**
> >
> > ### Further Responses
> >
> > > Two of the three proposed environments are solved already within a moderate amount of training steps with a recurrent agent.
> >
> > A major strength of our environments is scalability. We show varying difficulty levels in Mortar Mayhem and Mystery Path. Mortar Mayhem can be even further scaled. Especially, if the delays between observing and executing commands are sampled. From the results of DM Ballet we already know that sampling those delays degrades the performance of a recurrent agent. Also, it is beneficial to show that easier environment instances can be solved. Otherwise this would be a highly unexpected result, which would raise questions to bugs in the environment or in the agent. To ensure that these questions are not raised for Searing Spotlights, we ran extensive experiments like the ones on the modified Bossfight environment.
> >
> > > Why are only 10 novel seeds chosen for evaluation, but repeated 5 times?
> >
> > PPO features a stochastic policy. Therefore, repeating seeds during evaluation is advisable for a clean evaluation.
> >
> > > The authors claim that DM Alchemy, Crafter, or ObstacleTower require non-trivial additional components. What are those?
> >
> > This context is beyond the scope of our submitted work. But if you trace down some of the references you can find examples such as:
> > - DM Alchemy -> e.g. Auxiliary losses [1]
> > - Crafter -> e.g. Model-based approaches  [2]
> > - Obstacle Tower -> e.g. Domain Knowledge [3]
> >
> > > Why is open-sourceness explicitly stated as requirement?
> >
> > The already known advantages of open source software lead to inclusive research and not exclusive research. We contribute a benchmark and do not run a competition where every measure possible is taken to ensure the integrity of third-party results.
> >
> > > Why would HPC facilities not support dependencies such as xvfb, or EGL?
> >
> > We experienced an HPC administration that refused to support those dependencies. The more dependencies a software relies on, the less accessible the software is. Inaccessible tools hinder research progress.
> >
> > > Has HELM been properly tuned?
> >
> > Due to HELM's poor wall time efficiency, the hyperparameter search is limited. This is a weakness of HELM that has not been reported by the original authors. This is a minor insight that our paper contributes.
> >
> > > Why are training and test seeds for MM shown in one plot in Fig 5a, but in separate plots for MP in Fig 5b and 5c?
> >
> > As already written in the caption of Figure 5, the training and generalization performance in Mortar Mayhem are nearly identical. This way, we were able to save space for further important content.
> >
> > > [...] Spotlights are actually bad as there is no negative reward for losing health.
> >
> > The agent has two time budgets: max episode length and its remaining health points. Prematurely terminating the episode because of loosing all health points is bad for the agent as it is motivated to maximize the discounted return. This fact does not require negative rewards in this case. Also, we experimented with such a negative reward function that did not yield any benefit.
> >
> > At this point (comments are limited to 5k characters), we prune our response and refer to the to-be-released revision.
> >
> > [1] Wang et al. Alchemy: A benchmark and analysis toolkit for meta-reinforcement learning agents. Neurips 2021.
> > [2] Danijar Hafner. Benchmarking the Spectrum of Agent Capabilities. ICLR 2022.
> > [3] Pleines et al. 2020. Obstacle Tower Without Human Demonstrations: How Far a Deep Feed-Forward Network Goes with Reinforcement Learning. [2020 IEEE Conference on Games (CoG)](https://ieeexplore.ieee.org/xpl/conhome/9222389/proceeding).

---

> > > ### Author Response · Authors · 2022-12-07
> > > **Response #3**
> > >
> > > Thanks for your return and that you value our revision!
> > >
> > > > Regarding publicly available baseline implementations [...]
> > >
> > > The HELM authors did not publish their source on GTrXL, training TrXL from scratch, and fine tuning TrXL.
> > >
> > > > The MortarMayhem environment can also be turned into a supervised learning problem.
> > >
> > > You are also pointing out that DMBallet is not equivalent to Mortar Mayhem. We do not claim that our environments make DMBall obsolete. We claim that we have intriguing challenges that complement the current landscape. We argue that DMBallet does not rely on frequent memory interactions.
> > >
> > > > Crafter should be listed in Table 1
> > >
> > > Crafter is designed for multiple challenges as denoted in their paper: Exploration, Generalization, Reusable Skills, Credit assignment, Memory, Representation, and Survival.
> > >
> > > > MysteryPath indicate, that PPO can constantly infer up to 2 actions (Fig. 4a).
> > >
> > > Fig. 4a is not concerned with Mystery Path.
> > >
> > > > there is evidence that shows an [...] LSTM is incapable of solving DMBallet with only 2 dancers [...]
> > >
> > > You solely compare MMGrid to DMBallet. We propose the entire task of Mortar Mayhem which is a profound challenge.
> > >
> > > > I would suggest evaluation of more than 30 novel seeds [...]
> > >
> > > Expensiveness is relative. HELM has poor wall-time efficiency at inference. Evaluating one environment on all baselines takes us more than 24 hours. The only major changes in results were observed on novel seeds concerning Mystery Path. Therefore not all results are “drastically” changed. None of the updated results contradict our claim that MM, MP and Searing Spotlights are profound benchmarks. We evaluated Mystery Path Grid on 50 seeds. There is no significant change in results https://i.imgur.com/c5fWMjm.png .
> > >
> > > > Proper tuning of hyperparameters
> > >
> > > Which hyperparameters should be tuned that will lead to results contradicting the utility of our proposed environments? What is your reasoning? Searing Spotlights experienced a vast amount of tuning. Without evidence, this point may not be very convincing.
> > >
> > > > Claim that cumulative reward is not an appropriate measure for task completion
> > >
> > > We described that the mean cumulative reward in Procgen Miner is not helpful to assess the agent’s ability to solve the entire task. We do not claim that the cumulative reward is generally a bad measure.

---

> > ### Comment · Reviewer_e7qG · 2022-11-25
> > **Response to Response (2/2)**
> >
> > ### Additional points:
> >
> > Regarding publicly available baseline implementations, a possibility to include GTrXL as a baseline would be to train it with PPO, as GRU-PPO.
> > The same was done in prior work as well [2].
> >
> > The MortarMayhem environment can also be turned into a supervised learning problem by the exact same markovization as the authors use as an example for DMBallet.
> > Markovization in that case would require storing the sequence of commands observed at the beginning and the sequence of commands executed thus far, resulting in a supervised learning problem.
> >
> > The authors state that the Crafter environment [3] depends on non-trivial components.
> > In fact, Crafter does not require model-based approaches, and a recurrent PPO implementation outperforms model-based approaches [4].
> > Therefore Crafter should be listed in Table 1 as well.
> >
> > The authors made some contradictive statements in their response, such as:
> >
> > 	When considering Memory Gym's grid environments, the agent cannot infer its past action from its rotation. Based on its momentary observation, it cannot determine whether it rotated left, rotated right, moved forward, or did nothing at all. In the case of the multi-discrete action space and if the agent did not choose the no-op action, the agent could mostly infer its past action from its rotation.
> >
> > If the agent moved left in its last action, the agent will always face left which is represented visually, is that correct?
> > If that is not the case it should be clarified.
> > Also, the results for PPO on MysteryPath indicate, that PPO can constantly infer up to 2 actions (Fig. 4a).
> >
> > Several points of my review have neither been addressed in the author's response nor in the revised paper, for example:
> >
> > - Definition of "working memory mechanism"
> > - Claim that cumulative reward is not an appropriate measure for task completion (highest reward possible accomplished means task completed)
> > - Not clear whether agents condition on observations, or actions, or both
> > - "Uncertainty of an environment" -> Do the authors mean stochasticity?
> > - Interpretation on why some methods plateau when using continuous movement of agent
> > - Proper tuning of hyperparameters
> >
> > In light of the above-mentioned points, I can not increase my score for the current state of the manuscript.
> >
> > [1] Towards mental time travel: a hierarchical memory for reinforcement learning agents, Lampinen et al., NeurIPS 2021
> > [2] History Compression via Language Models for Reinforcement Learning, Paischer et al., ICML 2022
> > [3] Benchmarking the Spectrum of Agent Capabilities, Hafner et al., ICLR 2022
> > [4] Learning to Generalize with Object-centric Agents in the Open World Survival Game Crafter, Stanic et al., 2022, https://arxiv.org/abs/2208.03374

---

> > ### Comment · Reviewer_e7qG · 2022-11-25
> > **Response to Response (1/2)**
> >
> > Thank you for the detailed response and revision of the paper.
> >
> > As the authors point out in their general response, by adapting the evaluation protocol, some results drastically changed (before, performance on novel seeds even exceeded performance on training seeds for PPO on Searing Spotlights).
> > I appreciate the addition of the frame stacking baseline as a sanity check for longer-term dependencies and the clarification about environment specifics, such as the computation of episode lengths.
> >
> > Still, my main concerns have not been addressed by the authors:
> >
> > ### Novelty and Relevance:
> >
> > I acknowledge that MortarMayhem is an extension to DMBallet, however, there is evidence that shows an LSTM is incapable of solving DMBallet with only 2 dancers [1], while MMGrid is solved with a much lower training budget.
> > This raises the question, whether MMGrid is harder to solve than DMBallet, given that the agent needs to remember and discriminate between multiple dances in DMBallet, while it only needs to remember a single sequence of actions in MMGrid.
> > Furthermore, I would suggest evaluation of more than 30 novel seeds on MysterPath as it is not computationally expensive (compared to training) and the increased number of seeds compared to the first version already resulted in a major change in the presented results.
> >
> > Generally, the authors claim the purpose of the proposed benchmark is neither to evaluate robustness to noise, nor memory capacity.
> > However, their environments are designed to test for both (MM and MP for memory capacity, SS for robustness to noise).
> > Scaling the length of commands in MM, or the length of the path in MP requires storing more information (=memory capacity).
> > The authors themselves state that spotlights are actually noise:
> >
> > 	... the recurrent agent seems severely hurt by the perturbations inflicted by the randomly moving spotlights, which can be considered noise.
> >
> > Thus, SS evaluates robustness to noise.
> > In fact, the required information to remember in SS is easily remembered by GRU-PPO:
> >
> > 	In this case, if the global light is initially turned on (perfect information) and dimmed until off during the first few steps, the agent rapidly collects the coin and uses the exit.
> >
> > Moreover, the benchmark provides several degrees of freedom for scaling difficulty but lacks a unified protocol on how to meaningfully scale environments to enable proper comparison of different algorithms.
> > I also strongly disagree with the authors that the ability to exploit visual artefacts is desirable when the aim of the benchmark is to evaluate for memory.
> > In that case, it is actually a counterfactual that may result in misinterpretation of reported results.
> > Also, some design choices, i.e. last action as part of observation in SS, visual feedback instead of negative reward, multi-discrete action space, etc. are not justified.
> >
> > DMBallet and Procgen Memory Distribution fulfil all points of their desiderata.
> > With the above-mentioned evidence on LSTM on DMBallet, there is no clear advantage of MortarMayhem over DMBallet.
> > With respect to Procgen Memory the authors state:
> >
> > 	... while doubtlessly, a memory-based agent should be more efficient concerning the number of steps needed to solve the entire task.
> >
> > If memory-based agents should be more efficient, is it not still a valid benchmark for evaluating memory-based approaches?

---

### Author Response · Authors · 2022-11-18
**Revision Completed**

Dear Reviewers,

Thank you for the reviews, your insightful feedback, and hard work!
We found lots of value and inspiration.
Given the limited time of the revision, we tried our best to consider as much feedback as possible.
Hopefully our effort in improving this work finds value and appreciation.
These are the most apparent revisions:

### Major Revisions
- All experiments are evaluated by 30 environment seeds instead of 10
	- 10 seeds were insufficient on evaluating Mystery Path Grid (Fig. 4b)
	- Coincidentally, 5 training seeds generated straight paths at the border of the environment allowing for naive policies to succeed
	- Adding more seeds mitigates this issue (those 5 seeds were still included, i.e. we did not cherry pick seeds)
- Added HELM results on Searing Spotlights (Fig. 4c)
	- HELM performs best, but still reaches only a success rate of 0.3
- Added Frame Stacking baseline results (Fig. 4)
	- Stack of 4 RGB Frames
	- Stack of 16 Grayscale Frames
- Appendix D describes GRU-PPO (in great detail) and the other baselines
- Quantification of episode lengths (Appendix C.2)
	- Added equations to calculate the min and max episode length in Mortar Mayhem
	- Specified the episode lengths for all concerned environment variants
- Reworked the future work content of the conclusion


### Minor Revisions
- The term memory is further narrowed in the introduction
- Performance metrics of the results are now defined (Sec. 4)
- Added a figure (Appendix C.1) to illustrate the multi-discrete and the discrete action space
- Fixed Fig. 1 a)
- Cited Long Range Arena (Section 2.5)

---

### Decision · Program_Chairs · 2023-01-20

**Decision:**

Accept: poster

**Justification For Why Not Higher Score:**

There are some reviewers that are on the fence about the acceptance of the paper. Therefore, I could not give a better score.

**Justification For Why Not Lower Score:**

The work has many benefits for researchers from small labs.

**Metareview: Summary, Strengths And Weaknesses:**

The paper proposes a novel benchmark called Memory GYM to evaluate memory-based agents.

This paper has raised a significant amount of discussions among the authors, reviewers, and the AC. While there are mixed opinions about the usefulness of the proposed benchmark. After the discussion between the reviewers and AC, I vote for the acceptance of the paper conditional to the below points to be addressed. I tend to accept the paper mainly due to its benefit for academic labs with limited compute capacity. The benchmark enables practitioners to test various models in a contained set of tasks which could enable new ideas from small labs.

Major Concerns to be addressed:

- Addressing detailed comments of Reviewer e7qG: Reviewer e7qG provided great insights and advice on matters that must be addressed in your potential camera-ready version. In particular, a summary of the discussions you had during the discussion period must be reflected in the conclusions, discussions, scope, and limitation sections of your paper.
- Addressing the remaining concerns of Reviewer Arip: In particular, authors must tone down the claims throughout the paper and precisely reflect exactly what challenges (discrete spaces + sequential decision-making) have been explored. This must be reflected in the Abstract, intro, and the rest of the paper.
- The related works suggested by Reviewer aYs5, on candidate models that could be used, should be included in the revised version.

**Note From Pc:**

if the above contains the word "oral" or "spotlight" please see: "oral" presentation means -> notable-top-5% and "spotlight" means -> notable-top-25%. As stated in our emails, we are disassociating presentation type from AC recommendations

**Summary Of Ac-Reviewer Meeting:**

We discussed this paper in 2 different meetings with different participating reviewers. Here are the meeting minutes:

- The scope of the memory GYM is very limited as it does not include continuous-time action spaces.
- The authors did a great job during the discussion period addressing reviewers' concerns.
- Although the work is limited and narrow in scope, it would certainly benefit labs with limited access to compute resources to get hands-on and solve fundamental problems at the intersection of memory and learning.
- The novelty is limited as it builds on top of other work.
- Discussions of some potential sequence modeling candidates are missing.

All points have been taken into consideration as reflected in the detailed meta-review outlined above.